

# Examining the sustainability and development challenge in agricultural-forest frontiers of the Amazon Basin through the eyes of locals

Irene Blanco-Gutiérrez[1,2], Rhys Manners[3], Consuelo Varela-Ortega[1,2], Ana Tarquis[1,4], Lucieta G. Martorano[5], Marisol Toledo[6]

[1] CEIGRAM, Universidad Politécnica de Madrid (UPM), C/ Senda del Rey 13, 28040, Madrid, Spain
[2] Department of Agricultural Economics, Statistics and Business Management, ETSIAAB, Universidad Politécnica de Madrid (UPM), Av. Complutense s/n, 28040 Madrid, Spain
[3] International Institute of Tropical Agriculture (IITA), KG 563, Kigali, Rwanda
[4] Department of Applied Mathematics, ETSIAAB, Universidad Politécnica de Madrid (UPM), Av. Complutense s/n, 28040 Madrid, Spain
[5] EMBRAPA Eastern Amazon, Trav. Dr. Enéas Pinheiro s/n° Caixa Postal, 48, CEP 66.095-100 Belém, Brazil
[6] Instituto Boliviano de Investigación Forestal, Km 9 Carretera al Norte, El Vallecito, Santa Cruz de la Sierra, Bolivia

*Correspondence to*: Irene Blanco-Gutiérrez (irene.blanco@upm.es)

**Abstract**

The Amazon basin is the world's largest rainforest and the most biologically diverse place on Earth. Despite the critical importance of this region, Amazon forests continue inexorably to be degraded and deforested for various reasons, mainly a consequence of agricultural expansion. The development of novel policy strategies that provide balanced solutions, associating economic growth and environmental protection, is still challenging, largely because the perspective of those most affected- local stakeholders- is often ignored. Participatory Fuzzy Cognitive Mapping (FCM) was implemented to examine stakeholder perceptions towards the sustainable development of two agricultural-forest frontier areas in the Bolivian and Brazilian Amazon. A series of development scenarios and a climate change scenario were explored and applied to stakeholder derived FCM. Stakeholders in both regions perceived landscapes of socio-economic impoverishment and environmental degradation driven by governmental and institutional deficiencies. Under such abject conditions, governance and well-integrated social and technological strategies offered socio-economic development, environmental conservation, and resilience to climatic changes. The results suggest the benefits of a new type of thinking for development strategies in the Amazon basin, and that continued application of traditional development policies reduce the resilience of the Amazon to climate change, whilst limiting socio-economic development and environmental conservation.

**Key words:** agriculture expansion, deforestation, stakeholder perception, fuzzy cognitive mapping, sustainable development strategies, Amazon basin.



## 1 Introduction

The Amazon basin is one of the richest terrestrial biodiversity hotspots, a globally significant carbon sink, a leading supplier of agricultural commodities (Lapola et al., 2014), and home to millions (INE, 2014). The contemporary basin is the product of prolonged socio-natural interactions (Ioris, 2016), with environmental destruction and degradation increasingly

commonplace (Foley et al., 2007). This degradation constitutes a threat to biodiversity (Haddad et al., 2015), ecosystem service provision (Brandon, 2014), and climate change mitigation (Harris et al., 2012). These environmental changes have been driven by agricultural and other extractive activities (Hosonuma et al. 2012), which have expanded to profit from the basin's resources (Weinhold et al., 2013). The changes are so widespread that Davidson et al. (2012) describe the basin as a region in transition. Furthermore, the basin is threatened by climate change, with temperatures having already increased at

least 1°C since the 19[th] century (Victoria et al., 1998). Continued environmental degradation and climatic changes could increase regional weather pattern variability (Spracklen and Garcia-Carreras, 2015) and threaten both biodiversity (Phillips et al., 2009) and agricultural activities (Oliviera et al., 2013). Further, future scenarios suggest a continuation of this precarious state (e.g. Folhes et al., 2015; Tejada et al., 2016), with Lenton (2011) proposing that tipping points could be reached.

Activities responsible for the current state of the basin have, in many cases, been supported by policies to encourage rural development (Diversi, 2014), and in some instances have catalysed socio-economic improvements (Le Tourneau et al., 2013). However, the long-term developmental benefits of such activities are unclear (Rodrigues et al. 2009; Celentano et al, 2012; Weinhold et al., 2015), with Ioris (2016) positing that developmental improvements from extracionist activities are far from universal and are overshadowed by long-term environmental costs. Policies that solely concentrate upon agricultural

intensification and resource extraction may provide only marginal economic gains (Ioris, 2016), and may be responsible for social and environmental negative effects in a long-term perspective (Weinhold et al., 2015). Conversely, conservation policies have been implicated as drivers of negative socio-economic impacts (Chomitz, 2007; Carr, 2009; Guedes et al., 2014). These findings point toward the trade-offs in rural development objectives (McNeil et al., 2012), which increasingly focus upon socio-economic development through extracionist activities, or environmental conservation that excludes them.

This dichotomy has dominated the political and developmental discourse of the Amazon for decades, with Nobre et al. (2016) suggesting they represent the basin's established development model.

The bleak state and outlook of the Amazon basin, along with the limitations of the entrenched development policies, beg the question as to whether other options exist to transition the basin towards a sustainable, less conflict-ridden state. Nobre et al. (2016) promote a "third-way", driven by investment in technical and social capital, catalysing a localised industrial

revolution. Guedes et al. (2014) offer that increased access to technical assistance may permit communities to develop more sustainable livelihoods, converting natural capital to social capital. Lapola et al. (2014) infer that technological improvements along with sustainable land management could drive a sustainable shift in land use in the Brazilian Amazon. A further possibility may lie in the results of recent analyses (e.g. Weinhold et al., 2015; Caviglia-Harris et al., 2016). These





analyses suggest that socio-economic development in forest frontier regions of Brazil has uncoupled from environmental exploitation and degradation, due to policy development and implementation (Lapola et al., 2014; Caviglia-Harris et al., 2016). Tritsch and Arvor (2016) suggest that recent improved governance structures have begun to address competing rural development goals. Godfray et al. (2011) and Newton et al. (2013) advocate that governance and institutional improvements

could provide a balance between conservation, development, and climate change mitigation. The implementation of such reforms, or similar strategies could offer an interesting discussion point to reassess the emphasis of rural development policies. However, consideration of novel strategies would be reliant upon modelling and testing, offering scope for scenario development and application. The development of such scenarios could aid in quantifying the impacts of potential strategies in improving factors within the three main rural development dimensions, social, economic, and environmental, whilst

simultaneously mitigating climate change.

However, in analysing the Amazon basin, development strategies and scenario development, it is easy to ignore the perspective of those most likely to be affected- local stakeholders. Local perspectives can be drowned out by the largely expert derived knowledge base that dominates the understanding of the Amazon basin. Stakeholder involvement can provide new interpretations to previously studied problems, improve the understanding of complex situations, reduce unforeseen

consequences of policy implementation, and empower local communities (Folhes et al., 2015; Olazabal and Pascual, 2016). Application of scenarios to stakeholder derived information may broaden the understanding of localised issues within the basin and highlight the effectiveness of traditional and novel development strategies in addressing such issues. A number of methods are available to incorporate stakeholder perspectives into such analyses (e.g. Verburg et al., 2014), including Fuzzy Cognitive Mapping (FCM). FCM involves the development of a visual representation (map) of perceptions of a given system

or situation (Kok, 2009) and permits the application of scenarios to these maps (e.g. Vasslides and Jensen, 2016).

Using stakeholder derived information collected from workshops performed in forest frontier communities of the Bolivian and Brazilian Amazon (the province of Guarayos in Bolivia and the Tapajós National Forest in Brazil), this paper aims to identify how such communities perceive the present state of their region using FCM. Further, this analysis will apply development and climate scenarios to these cognitive maps, analysing how each region reacts to the sustainability and

development challenge, changing socio-economic, political, and climatic conditions.

## 2 Methodology

### 2.1 Description of the study area

The Amazon basin is the largest tropical rainforest in the world. It covers an area of approximately 6 million km$^2$, extends over eight South American countries, and consists of wide mosaic of ecosystem and vegetation types. Given the size of the



region, two study sites have been selected in the framework of the ROBIN[1] project. Firstly, the Province of Guarayos (20,029 km$^2$, covering the municipalities of Ascensión de Guarayos, El Puente, and Urubichá), in the northwest corner of the Department of Santa Cruz in lowland Bolivia; the second, the Tapajós National Forest (5,449 km$^2$) bound by the Tapajós River, the Cupari River, and the Santarém–Cuiabá highway (BR-163), in the western part of the State of Pará
(municipalities of Belterra, Placas, Rurópolis and Aveiro), in northern Brazil (Fig. 1).

**Figure 1 here**

Both study sites provide representative examples of the threats that endanger the Amazon basin. Despite this, these threats
are highly conditioned by the specific characteristics for each region, which offers an interesting perspective for comparison. The Province of Guarayos (henceforth Guarayos) is located at the southernmost extent of Amazonian rainforest in Bolivia, in the transition zone between the humid Amazon forest and the dry Chiquitano forest. The climate is tropical, with a mean annual temperature and precipitation of approximately 22°C and 1600 mm. This region, like all southern Amazon regions, is prone to changes in precipitation and is expected to be most affected by rainfall declines caused by climate change (Malhi et
al., 2008). In Guarayos, half of the territory is covered by natural forests. It hosts protected forest areas, such as the 'Reserva Nacional de Vida Silvestre Ríos Blanco y Negro' created in 1990, which are hugely important in terms of biological diversity. In the vicinity of theses protected areas lives the Guarayos indigenous community (a branch of the Guaraní), whose livelihoods depend on fishing, hunting, and gathering fruit, as well as the cultivation of rice, pineapples, bananas, manioc, and other crops. The extraction of wood is limited. In spite of significant efforts to promote formal sustainable forest
management at the community and industrial level, only informal timber networks have been developed (Albornoz et al., 2008). Since 1996, land is collectively owned and managed by the Guarayos through a 'community land of origin' (TCO, by its Spanish acronym), which has contributed to the sustainable conservation and utilisation of forests. However, the legal uncertainty surrounding the system of land tenure in Bolivia, coupled with the increasingly frequent arrival of outside investors in the area, mainly large-scale farm operators, have resulted in highly conflicted situations, with illegal
appropriation of TCO lands and environmental degradation (deforestation, contamination, habitat destruction, soil degradation, etc.) (Killeen et al., 2008; Stavenhagen, 2009). Between 2001 and 2012, the population of Guarayos almost doubled to 48,301 inhabitants. Currently, agriculture is the primary economic activity, with almost 50% of the working population employed in agricultural activities (INE, 2011). Of the total arable land (4% of the province's land surface area), soya dominates both winter and summer cultivation, followed by sunflower, maize, rice, and sorghum (INE, 2015). In
general, deforestation and the expansion of the agricultural frontier in Bolivia has been less well studied than in Brazil, probably due to its relatively recent development and it being concentrated in the Department of Santa Cruz to the east

---

[1] The research project ROBIN (The Role of Biodiversity in Climate change Mitigation) (2011-2015), funded by the European Union Seventh Framework Programme under grant agreement No 283093, aims at quantifying interactions between terrestrial biodiversity, land use and climate change potential in tropical Latin America. More information can be found at https://cordis.europa.eu/project/rcn/100815/reporting/.





(Pacheco, 2006; Killeen et al., 2008). The socio-ecological implications of the expansion of agricultural frontier in this region are huge, with increasing efforts being made to study this part of the Amazon basin.

The Tapajós National Forest (henceforth Tapajós) is located at the heart of the Amazonian rain forest in Brazil. In this region, the climate is humid tropical; the mean temperature is 26°C and annual precipitation averages approximately 1,820
mm. The dry season lasts roughly two months, falling between August and October, with rainfall of < 60 mm month $^{-1}$ (IBAMA, 2004). The natural vegetation is dominated by dense moist and wet forest types with emergent trees (Dubois, 1976). Tapajós is the second oldest conservation unit in the Brazilian Amazon. It has been protected since 1974, when it was officially designated as a 'National Park', and classed as an IUCN category VI protected area (protected area with sustainable use of forest resources and scientific research) (IBAMA, 2004).

The area is home to 5,000 'ribeirinhos' (traditional South American populations living near rivers), distributed across 16 communities mostly along the Tapajós River. These communities are well organised and have historically been very active in governance processes. During the 30-year period (1980-2010), the traditional riverine population held an important resistance movement to avoid eviction and gain land tenure and resource rights. This movement was a pioneer in Brazil and led to a commercial community forest management system that has attracted both national and international attention
(Bicalho and Hoefle, 2015). Despite this, these communities face difficult living conditions, with poor access to services (education, health, etc.). Logging is the main source of employment and revenue for the population, who subsist on very low incomes subsidised by small-scale subsistence farming activities (manioc, beans, and corn), fishing, hunting, and non-logging activities (eco-tourism and the sale of wood-latex-leather handicrafts). As a result, most residents are dependent on government transfer payments (Hoefle, 2016). Although forestry exploitation in Tapajós is mostly carried out in a
sustainable way, by the local population, growing concerns regarding the conservation of protected areas have recently emerged. External pressures on these protected areas are increasingly being applied by private forestry companies to acquire concessions, expansion of intensive agriculture and cattle grazing areas coming mainly from the neighbouring Cerrado, and the development of infrastructure (highways and dams) for the acceleration of growth (Gibbs et al., 2015; Verburg, 2014). The bordering Santarém-Cuiabá (BR-163) and Transamazonian (RB-230) highways, planned to be reconstructed, are
considered major corridors of deforestation as they stimulate migration and exportation of livestock, soybean, minerals (gold), and forestry products via the Amazon River (Fearnside, 2007). Beyond this, the Tapajós River is at the centre of some of the most recent and dynamic hydroelectric development activity in Brazil (Fearnside, 2015). In 2012, the Brazilian government approved a law (No. 12,678) to enable the construction of the São Luiz do Tapajós mega dam, which would have reduced the geographical limits of the Tapajós National Park by 11,990 ha. This particular project was highly criticised
and finally cancelled in 2016, but similar ones are still planned and threaten the study area.

## 2.2 Participatory development of FCMs

FCM is attributed to Kosko (1986) who provided the fuzziness to earlier cognitive mapping techniques (Tolman, 1948; Axelrod, 1976). Maps developed from FCM visualise components and their causal relationships within a system (Kok, 2009)



as perceived by an individual, or group. This mapping can be developed through participatory interviews or workshops, where components (nodes, concepts or vertices) representing features of the system are identified, and causal relationships (links, connections or arcs) between them are defined through weighted and meaningful directed linkages (Gray et al., 2015). These relationships range from -1 to +1 (Özesmi and Özesmi, 2004) and define the scale of influence (positive or negative)

that one component has upon another.

The causal networks developed from FCM have considerable flexibility for analysis in a range of fields (e.g. Papageorgiou et al., 2013) and support scenario development (e.g. Kok, 2009). The methodology can incorporate multiple stakeholders' perspectives and knowledge (Gray et al., 2015) through combination of multiple maps into one 'community' map (Fairweather, 2010) or development of a single map by a group of stakeholders (Varela-Ortega et al., 2014), aggregating and

incorporating distinct perspectives of different groups into a single vision. Participatory development of FCMs can improve communication through the development of an open, neutral, and informal forum for participants to give their opinions. The FCM methodology can incorporate both measurable (e.g. deforestation) and qualitative concepts (e.g. awareness of environmental problems). FCM can provide useful output for data scare problems or in areas where data it is difficult to obtain and can be complementary to quantitative models (Olazabal and Pascual, 2016). The results of FCM are semi-

quantitative and can only be interpreted relative to other values within the system (Kok, 2009).

In this study, we use FCMs to visualise the perceptions of local stakeholders concerning the direct or indirect interactions of variables that influence the state of the local environments in both Guarayos and Tapajós. The steps implemented as part of the methodology are illustrated in Fig. 2.

**Figure 2 here**

In each of the case studies, two stakeholder workshops were held within the framework of the ROBIN project. In the first, we facilitated two focus groups, consisting of diverse groups of stakeholders providing a heterogeneous perspective (Table 1).

**Table 1 here**

Each focus group developed its own FCM. Participants were invited to offer their perspectives on the present state of the environment in the region and what they considered to be the key features and processes inherent to it. Participants discussed

and identified the components, causal component connections, and the weights of these connections in the development of the maps.

After the first stakeholder workshop and following Ösezmi and Ösezmi (2004), the two group maps from each case study were combined into one 'Case Study FCM'. As part of the combination process, components identified as representing similar features were merged, where possible. However, in combining components, conflicting connections were identified,





normally involving the wording "Lack of…" In these cases, and following Vasslides and Jensen (2016), wording of the more prevalent component was kept, and connection weights were inverted appropriately.

The combined FCM was presented in the second workshop for enrichment, validation, and interpretation. Once the 'Case Study FCM' was agreed, a discussion on possible futures and sustainable strategies was held, serving as input for scenario development and simulation. To ensure continuity, care was taken that similar stakeholders (or stakeholder groups) were present in the second workshop.

## 2.3 FCM analysis

The two 'Case Study FCMs' were analysed following Reckien (2014) and Olazabal and Pascual (2016) considering their structure, dynamics, and the impacts of scenarios on their dynamics.

### 2.3.1 Structural metrics

As FCM are considered complex networks, the structural metrics here used to analyse them are complex network parameters commonly applied in the literature (see Table 2). Further, we also include two novel metrics for the measurement of centrality in FCM analysis: page rank (PR) and betweeness (Bw). In the two networks analysed (FCM of Guarayos and Tapajós) the ties among nodes have weights assigned to them, therefore the FCM are considered weighted networks and the centrality measures are weighted as well.

**Table 2 here**

Bw was first introduced by Freeman (1977) to quantify the control that an individual can achieve on the communication between other humans in a social network. PR was named after Larry Page (Page, 1999), one of the founders of Google, and is used by Google Search to rank websites in their search engine results. While Bw measures the influence of a node within a network by calculating the number of times a node acts as an intermediary along the shortest path between two other nodes, PR calculates the probability of visiting each node if we were randomly 'surfing' the net.

### 2.3.2 Dynamic analysis

Besides the structural metrics of Table 2, the dynamic behaviour of the maps was also analysed to gain an insight into how components interact with each other, over various iterations (Gray et al., 2015). This analysis permitted comparison between the steady state values (Kosko, 1994) for each component, as well as the simulation of scenarios.

To calculate the steady state values and perform the dynamic analysis, each Case Study FCM was converted into an adjacency matrix, which was then multiplied by a state vector $A$ (Eq. 1) over various iterations ($k$). According to Kok (2009), this calculation results in four potential dynamic outcomes: components return to zero, components continuously increase/ decrease, components continuously cycle, and components stabilise at a fixed value.





$$A_i^{(k+1)} = f\left(A_i^{(k)} + \sum_{\substack{j=1 \\ j\neq i}}^{N} A_j^{(k)} w_{ji}\right) \qquad (1)$$

Where $A_i^{(k+1)}$ is the value of the component $C_i$ at iteration $k$ +1; $A_i^{(k)}$ is the value of component $C_i$ at iteration $k$; $A_j^{(k)}$ is the

value of the component $C_j$ at iteration $k$; and $w_{ij}$ is the weight of the connection between components $C_i$ and $C_j$.

The state vector $A$ initially sets values for all components to 1 (Olazabal and Pascual, 2016), assuming all components are

equally important and is multiplied against the adjacency matrix. The resultant vector is transformed to a logistic expression

$f$, binding values between 0 and 1 (Kosko, 1986). This output vector is once again multiplied against the adjacency matrix,

producing bound results between 0 and 1. This process is repeated until the dynamic outcome becomes evident, usually after

20-30 iterations (Kok, 2009).

Output (steady state) values close to 0 are representative of a strong decrease in the component, whereas values closer to 1

represent a strong increase (Reckien, 2014). The steady state values were interpreted as the current state of each component

within the system (map) and were used as a baseline for interpreting the impacts of the scenarios.

**2.4 Scenario development**

Five scenarios were developed to identify how Guarayos and Tapajós may react to the conditions of four development

strategies and to climate change (Table 3). The four development strategies were characterised to replicate the traditional

binary strategies applied in conflicting agricultural-forest frontier areas (environmental conservation, associated with

scenario 3 in Table 3; and agricultural development, scenario 4) (Nobre et al., 2016), with two others characterising

governance (scenario 1) and techno-social reforms (scenario 2). Further, a climate change scenario (scenario 5) was applied

to analyse the impact of climatic changes on the current state of the system. Although addressing similar concepts and

themes, the scenarios differ in their application and characterisation across the case studies due to differences in the mapped

systems' constituent structure.

To analyse the impacts of the scenarios, the same calculation as for the baseline (Eq. 1: Sect. 2.3.2) was performed.

However, unlike the baseline, where the state vector for each component was fixed at 1, the scenarios fixed the values of

certain components within state vector. Those components included within the scenario (Table 3) had their values fixed at a

set value between 0-1, depending upon the assumed impacts of the scenario on that component. All other components had

their values set to 0 within the state vector (Kosko, 1986). If for example 'Lack of Government Coordination' was included

within an improved governance strategy, its value would be set lower than that the steady state value of the baseline,

characterising the assumption that the strategy will reduce the impact of this component. The five scenarios, their

description, the components, and the fixed values are presented in Table 3.



**Table 3 here**

The output values for components under each scenario were then compared to their baseline values, with differences suggesting the relative impacts of each scenario. Further, the effects of the four development scenarios were also tested under the conditions of climate change scenario.

To determine the wider impacts of the scenarios on the system, cumulative impacts for each scenario were analysed. To do so, components were categorised as positive, negative, or neutral (Reckien, 2014; Olazabal and Pascual, 2016) (Supplementary Table S1). Categorisation of components was based upon the perception of the role that each component would have in developing more sustainable regions. Components were categorised to recognise the equal importance of a reduction in a negative component, as an increase in a positive one, when considering the cumulative impacts of the scenarios. As with Reckien (2014), an aggregated impact value was calculated as the sum of: increases in positive components and decreases in negative components (from baseline to scenario).

It should be noted that the output results of FCMs are semi-quantitative. As such, outcomes can only be used to determine impacts on components, relative to other components, rather than absolute changes (Özesmi and Özesmi, 2004; Kok, 2009). Impact comparisons can only be made within the system and cannot be compared with absolute indicator values (Reckien, 2014; Devisscher et al.,2016).

## 3 Results

### 3.1 Structure analysis of FCM

Analysis of the two Case-Study FCMs demonstrated structurally similar systems (Table 4), with divergent contents (Fig. 3, 4 and 5).

**Table 4 here**

The two maps have comparable component numbers and similar densities of 0.052 (Guarayos) and 0.048 (Tapajós). The density difference may suggest that stakeholders in Tapajós perceive greater causal relationships between components. According to Özesmi and Özesmi (2004) this may offer greater possibilities to elicit change within Tapajós, compared to Guarayos. The complexity of the Guarayos map (0.57) was almost double that of Tapajós (0.33), suggesting that Tapajós is a more hierarchical system (Özesmi and Özesmi, 2004), with more transmitting components. This hierarchical lean is reflected in the components of the Tapajós map (Fig. 5), dominated by political and institutional concepts and problems, whilst the Guarayos map (Fig. 4) appears more heterogeneous.



A first look at the results obtained in Bw and PR (Table 4) shows that the maximum Bw value in Guarayos is double than in Tapajós, 0.21 and 0.09 respectively, as we observed with Complexity. In both cases the highest Bw corresponds to Deforestation. Meanwhile PR maximum values are more similar in both case studies being  higher in Tapajós  than in Guarayos. Studying the values distribution for both metrics (Bw and PR) in percentage of components, it is possible to

compare both cases. With respect to Bw (Fig. 3a), the highest six values are quite differentiated from the rest, in Guarayos showing a range from 0.05 till 0.21. These correspond with ordinary components: Agricultural Expansion, Climate change, Illegal logging, Lower crop yields and Deforestation. In the case of Tapajós, there is only one differentiated value corresponding to Deforestation. With respect to PR (Fig. 3b), both cases present several differentiated values that are visualized in the network (Fig. 3 and 4) for a deeper analysis.

**Figure 3 here**
**Figure 4 here**
**Figure 5 here**

Both systems are dominated by environmental problems, with deforestation and biodiversity loss having the highest page rank value in Guarayos and Tapajós. It is also important to note, the importance of poverty and low crop yields in Guarayos and forest products value and population purchasing power in Tapajós. In Guarayos, deforestation is the most influential component (highest outdegree, see Table S2) driving climate change, soil erosion, and biodiversity loss (Figure 4), whereas in Tapajós deforestation is the most influenced component (highest indegree, see Table S3) affected by amongst others:

infrastructure projects, lack of public policy, and agricultural expansion (Fig. 5). In Tapajós, a lack of efficiency in policies for subsistence farmers enacts the greatest influence (highest outdegree, see Table S3), causing incomplete production chains, lack of technical capacity, and access to viable economic activities (Fig. 5). Components including contamination and biodiversity loss were found in both maps to have high indegrees (see Tables S2 and S3), suggesting their sensitivity to other components.

In Guarayos and Tapajós the aggregated page rank of the component groups was dominated by the environmental and economic groups, followed by political, social, and technical. In both maps, the environmental grouping is the most heavily influenced and sensitive group with the highest group indegree values. The components identified as transmitters (square components) were largely political and economic, mostly defined as ineffective or with negative connotations, with the use of words such as "Lack of..."or "Poor..." The influence of these components in driving the situation in both regions (Fig. 4

and 5) is supported by their outdegree values (Tables S2 and S3). The sensitivity of environmental components was once again demonstrated by the majority of receiver components (diamonds) being environmental.

Despite the differences in components within each map there was still overlap between them, with 15 of the 61 total components representing similar concepts (environmental degradation, worsening socio-economic situations, and poor



governance). This suggests that despite the maps being developed in distinct regions and with unique stakeholders, there is some continuity in the problems that afflict both regions and potentially the basin as a whole.

## 3.2 Dynamic analysis of FCM

### 3.2.1 Baseline situation

Dynamic analysis of the aggregated maps (Fig. 6 and 7) demonstrate significant overlap, despite the ~2000km that separate the case studies them. Both regions (Guarayos and Tapajós) are characterised by worsening environmental degradation and apparently bleak socio-economic opportunities for local communities, coupled with low institutional safeguards.

**Figure 6 here**

Figure 6 characterises Guarayos as a region where environmental degradation is high, facilitated by low (and declining) application of the forest law and poor (and worsening) compliance with land zoning, coupled with low socio-economic opportunities. The system is dominated by increasing contamination, deforestation, loss of biodiversity, soil erosion, fires, poverty and agricultural expansion.

**Figure 7 here**

The situation in Tapajós (Fig. 7) depicts a similarly degraded system, where environmental conditions are deteriorating, facilitated by limited economic opportunities, and poor environmental monitoring. Tapajós is dominated by loss of environmental services and biodiversity; and increasing contamination, deforestation, infrastructure projects, and agricultural expansion. Contrarily, socio-economic opportunities for locals are apparently diminishing with reducing value of forest products and limited access to viable economic activities. Further, monitoring of environmental degradation is inhibited by limited environmental monitoring.

### 3.2.2 Scenario outcomes

Figure 8 establishes the aggregate effects of the four development strategies and the climate scenario on the mapped system. The values for the components fixed within each scenario have not been included, to highlight the subsequent systemic impacts of changes to components fixed within each strategy.

**Figure 8 here**





The governance strategy was responsible for the greatest 'desired' change in both Guarayos and Tapajós, with the climate change scenario causing the biggest 'undesired' change. The techno-social and conservation strategies also resulted in desirable changes. However, application of the agricultural development strategy worsened the situation in both regions. Guarayos is more heavily influenced by climate change than Tapajós, which considering the page rank of climate change in

both systems (Fig. 4 and 5) may have been expected.

A more detailed description of the individual impacts of the scenarios on components in both systems is given below, with the extent of component changes shown in Supplementary Fig. 1 and 2. In general, implementation of these strategies results in greater changes to individual components in Guarayos than in Tapajós, which may be attributable to the higher density of the Guarayos map.

The *governance strategy* results in the greatest systemic relative changes and some of the greatest changes to individual components. This may demonstrate the integrated nature of governance components and their connectivity within both systems. The strategy encourages reductions in environmental degradation across the two systems including deforestation, logging, and forest fires. It also drives socio-economic improvements reducing poverty, increasing access to financial aid and viable economic alternatives, improving population purchasing power in Tapajós and reducing the inequality of benefits

in Guarayos. In Tapajós, it also elicits considerable improvements in the technical capacity of the region.

The *techno-social strategy* encourages a suite of positive changes to both systems, reducing environmentally degrading activities, whilst providing simultaneous economic development. In Guarayos poverty is reduced, along with reductions in contamination, deforestation, illegal hunting, and logging. The strategy provides similar reductions in environmental degradation in Tapajós, with large reductions in deforestation and fires, whilst increasing population purchasing power and

improving the value of forest products. Further, it also encourages greater social organisation and political participation, demonstrating a potentially beneficial unforeseen knock-on effect of such reforms.

The *conservation strategy* has limited impacts across the two systems, fomenting change only on environmental components. In Guarayos it reduces deforestation, whilst in Tapajós it reduces deforestation as well as other environmental degrading activities including; forest fires, logging, deforestation, and biodiversity loss.

The *agricultural development strategy* encourages substantial differences in the responses of the two systems. In Guarayos, crop yields improve with the expansion in both agriculture and grazing expansion, and results in reductions in poverty. Further, it also encourages positive environmental change with reduced illegal logging, hunting and fishing. However, in general environmental conditions worsen greatly with for example deforestation increasing, along with contamination, soil erosion, loss of biodiversity and destruction of pampas. In Tapajós, the rural development strategy results in no socio-

economic benefits, but encourages considerable environmental degradation with deforestation, forest fires, loss of environmental services and biodiversity and contamination all increasing.

The *climate change scenario* suggests that without immediate reforms to mitigate or adapt, the situation in Guarayos and Tapajós will worsen into the future.



Figure 9 demonstrates the sensitivity of the systems under each scenario, whilst experiencing continued climate change, with some scenarios demonstrating greater resilience than others.

**Figure 9 here**

Figure 9 reveals that the governance reforms (and to a lesser extent techno-social reforms) may provide the most effective and resilient means of instigating regional improvements, even under climate change. In Guarayos, the effect of climate change was so great that despite the conservation strategy the overall state worsened, compared with the baseline. In Tapajós, the impacts of climate change were still notable, but not to such an extreme extent as to further worsen the region.

In both Guarayos and Tapajós, the agricultural development strategy offered the least resilient development strategy.

## 4 Discussion

### 4.1 The Amazon as mapped by Stakeholders

The utility and flexibility of Fuzzy Cognitive Mapping to elicit a stakeholder-derived interpretation of the present state of two forest frontier regions of the Amazon basin has been demonstrated in this analysis. FCM afforded the combination of knowledge from regional experts and local community members, offering the opportunity to improve and enrich the understanding of these regions, whilst providing a low-resolution demonstration of their present state. We also outline the

potential to include novel network analysis metrics into parsing out the current situation of the Amazon. The highest values in PageRank and Betweeness are useful to detect the key components in the network. The use of FCM also facilitated the use of scenarios to analyse how these regions may react to development strategies, and climate change.

Despite the two maps reflecting systems on opposite sides of the Amazon basin, they yielded strikingly similar results. Stakeholders in both Bolivia and Brazil mapped systems plagued by environmental degradation, with social and governance

support structures absent or ineffective, inhibiting local community benefits. The perceived lack of effective governance is apparently incongruent to the contemporary literature, which suggests recent improvements in the governance model (World Bank, 2016). The presence of inequality, poverty, and deforestation are consistent with the paradox of poverty in resource rich systems (Ioris, 2016), with stakeholders appearing to characterise the same "...landscapes of impoverishment..." as Ioris (2016, p. 187). Stakeholders in both Bolivia and Brazil identified similar barriers to development, with poor governance and

conflicting policy measures inhibiting widespread socio-economic development, and hindering environmental conservation, supporting previous findings (Simmons et al., 2007). Further, the inconsequential nature of climate change for stakeholders in both cases was unexpected, considering its already noted impacts (Victoria et al., 1998) and potential future impacts (e.g. Spracklen and Garcia-Carreras, 2015). This unanticipated outcome may support the findings of Brondizio and Moran (2008),



who suggest that the memory of climatic changes is short-lived. However, Varela-Ortega (2014) found that stakeholders considered climate change a fundamental component in the future of both regions.

## 4.2 Affecting positive change in the Amazon

Implementation of the suite of scenarios affected substantial and variable changes. Governance and institutional reforms appear to offer the most effective means of transitioning Amazonian regions towards more sustainable 'desirable' states, even under the conditions of climate change. The positive effects of governance and institutional reforms are unsurprising considering the constraining effect (McNeil et al., 2012) that poor governance can have in inhibiting sustainable

development, with its effects well documented in the Amazon (e.g. Rodrigues-Filho et al., 2015). The results evidence the liberating effect that improving institutional capacity can have in instigating desirable social, economic, and environmental change. These multi-dimensional benefits apparently confirm the transversal nature of institutions and governance in the context of sustainable development (McNeil et al., 2012). The positive impacts of governance have precedence in the Amazon, where institutional and governance improvements have encouraged environmental conservation (Nepstad et al.,

2014; Tritsch and Arvor, 2016) and socio-economic development (Caviglia-Harris et al., 2016). Further, the literature widely supports the need for strong governance and institutions with Müller (2014), Verburg et al. (2014b), and Høiby and Zenteno-Hopp (2014) contending that the likelihood for long-term environmental conservation is slim under poor governance conditions. Lapola et al. (2014) promotes the need for policy enforcement and institutional support to encourage sustainable development, whilst Guedes et al. (2014) propose that pathways towards future environmental conservation can be founded

upon investments in local institutions.

Techno-social reforms also represent an alternative strategy, driving environmental protection, economic development, and social improvements. In Brazil, the difference in desired change between this strategy and governance reforms was minimal, suggesting its considerable potential. These results support the vision of Nobre et al. (2016), where rural development is encouraged through social and technological reforms, with both environmental and social components improving. The

implementation of this scenario suggests that investments in technical capacity building and social reforms may reverse the poverty traps (Reardon and Vosti, 1995) in which stakeholders mapped both regions appear to be locked. Investments in social and technical reforms may have wider unforeseen benefits, improving societal attitudes towards natural capital conservation (Salahodjaev, 2016), aiding in flattening environmental Kuznet´s curves (Tritsch and Arvor, 2016), and driving positive changes in agricultural methods (Assunção et al., 2013). Many of these points are suggested in the results of this

analysis. However, this strategy was admittedly found to be susceptible to climate change, more so than the institutional reforms.

Traditional developmental strategies relying upon conservation or extractionist policy implementation have driven trade-offs across the Amazon (Le Tourneau et al., 2013). The impacts of these binary choices can be stark, with decision makers having to make substantial compromises between environmental conservation and agricultural development (e.g. Manners



and Varela-Ortega, 2018). The application of the conservation strategy had limited system wide impacts, resulting in environmental improvements, but offering little opportunity for socio-economic development, potentially confining local communities to conditions of poverty and limited development. Further, implementation of such a narrow strategy was found to be particularly susceptible to climate change. The application of this strategy, or one similar, may have little chance of

providing sustainable rural development without concomitant offering of economic alternatives for locals, or the need for systems like Payments for Ecosystem Services to potentially alleviate poverty and encourage conservation (Pinho et al. 2014). Tejada et al. (2016) found that limiting future environmental degradation, specifically deforestation, in the Bolivian lowlands without offering new economic alternatives is unlikely.

The results also outline the negative effects of a strategy solely focussing upon agricultural development, with the long-term

benefits limited, especially under climate change. This strategy improved social factors like poverty and inequality (in Bolivia), but at a cost to local ecosystems in both Bolivia and Brazil. The outcomes of this scenario appear consistent with the literature, suggesting that purely agriculturally orientated strategies, without supporting policies may result in limited economic benefits for locals (Rodrigues et al. 2009; Ioris 2016) and some environmental costs (Weinhold et al., 2015). Further, these results appear not to demonstrate the uncoupling of agricultural development from environmental degradation

as identified in Brazil (Caviglia-Harris et al., 2016). However, focussing solely upon the local-scale economic and social benefits of such extractive strategies, as touched upon by Celentano et al. (2012), may ignore their wider national developmental benefits.

In summary, application of the two traditional scenarios for rural development (agricultural development and environmental conservation) demonstrate the trade-offs in their application and their ability to improve regional economic, social, and

environmental conditions. Development of new strategies concentrating upon governance and techno-social reforms could instigate positive shifts in the trajectory of these regions, even under the effects of climate change. However, moving from the modelled world to the real, where implementation of such strategies requires: consideration of social acceptability; likelihood of implementation; willingness of politicians and institutions to reform; coherence with current policy landscapes; and funding availability may result in complications. Despite improvements in governance across many Amazonian

countries in recent decades (World Bank, 2016), implementation of the governance reform may be challenging, especially under increasingly turbulent political landscapes, exemplified by Brazil. Further, potentially intangible (in the short-term) and time-consuming governance and institutional reforms may be unpalatable for voter conscious and electioneering administrations. Governments wanting to appear proactive in terms of rural development may consider other, more palpable options. The benefits of institutional reforms may only be reaped in the long-term, by which time governments may have

changed and the benefits of change lost for the implementing administration. This may highlight the space for market-based interventions to encourage more sustainable development (e.g Nepstad et al., 2014; Gibbs et al., 2015). Beyond this, strategies aimed at techno-social reforms may garner less systemic positive changes but offer more tangible actions for voters and governments alike, whilst fomenting positive change, even under worsening climatic conditions. However, the



financial implications of such reforms must be considered, with them likely requiring significant and long-term public or private investments. However, such funding is invariably scarce (Ferraro and Pattanayak, 2006).

**5 Conclusions**

The use of FCM to visualise the perceptions of stakeholders across the Amazon basin has shown that on both sides of the basin, landscapes of socio-economic impoverishment and environmental degradation are present, driven to varying degrees by governmental and institutional deficiencies. Even under such abject conditions, these processes have been modelled to be theoretically reversible through application of governance and well-integrated technical and social reform strategies. These

strategies were found to encourage positive regional changes even under the pressure of climatic change. However, what is apparent in both regions is that a continuation of the current rural development programmes cannot continue, with these results showing that concentration on only conservation or agricultural development policies would reduce the resilience of both regions to climate change, whilst also providing limited socio-economic development and continued environmental degradation.

**Author contribution**

The conceptualisation and methodology design were done by CVO (PI of Spain's research team in the ROBIN project) and IB. The development and implementation of the stakeholder workshops were done by IB, CVO, LGM and MT. AT carried

out the mathematical analysis of FCM. RM performed the simulations and supported the scenario development and processing of results. IB and RM prepared the manuscript with contributions from all co-authors.

**Competing interests**

The authors declare that they have no conflict of interest.

**Acknowledgements**

This investigation received funding from the research project ROBIN (The Role of Biodiversity in Climate change

Mitigation) (2011-2015) of the European Union Seventh Framework Programme under grant agreement No 283093- (https://cordis.europa.eu/project/rcn/100815/reporting/). Further financial support was also received from the Universidad Politécnica de Madrid through their Latin American Cooperation Programmes- Biodiversidad y bienestar humano ante el cambio global en áreas tropicales protegidas de América Latina (Project No AL13-PID-18) and Biodiversidad y cambio climático en la Amazonía: Perspectivas socio-económicas y ambientales (Project No AL14-PID-12). The authors are





indebted to the numerous stakeholders that have taken part in this research and to Dr. Paloma Esteve for their valuable comments throughout the development of the research.

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





**List of tables**

Table 1: Stakeholder workshops held in Guarayos (Bolivia) and Tapajós (Brazil).

| Case Study | Workshop | Nº Stakeholders | Stakeholder Groups Represented |
|---|---|---|---|
| Guarayos (Bolivia) | First: 30<sup>th</sup> January 2013 | 30 | Organisation Centre of Guarayo Native People (COPNAG), Forestry Services, Tropical and Agricultural Research Centre (CIAT), Arado Foundation, Farmers Federation, Indigenous Forestry Association, Rio Blanco and Rio Negro Wildlife Reserve, Guarayos Timber Association (AMAGUA), Authority and Social Control of Forest and Land (ABT), Guarayos Indigenous Women Centre (CEMIG), Development Area Program (PDA), Guarayo Cattle Association (AGUAGUA) and Ascensión Inter-Ethnicity Centre (CIEA) |
| | Second: 18<sup>th</sup> June 2014 | 27 | Autonomous Government of Santa Cruz (GDASC), Department of Protected Areas (DIAP), Department of Agriculture (SEDACRUZ), Indigenous Guarayos Women' Centre (CEMIG) Las Misiones Radio, Radio Guaguazuti, Central Organisation of Native Guarayo Villages (COPNAG), Department of Natural Resources (DIRENA), Indigenous Guarayos Forestry Asscoaition (IRARAI) and the Community Centre Urubichá (CECU) |
| Tapajós (Brazil) | First: 27th November 2013 | 23 | Ministry of Agriculture (MAPA), The Federal University of Western Pará (UFOPA), Chico Mendes Institute for Biodiversity Conservation (ICMBIO), Hope Foundation (IESPES), EMBRAPA Eastern Amazon, Tapajós Community Leaders, The Nature Conservancy (TNC) and Luiz de Quieroz College of Agriculture (ESALQ-USP) |
| | Second: 28th November 2013 | 26 | Ministry of Agriculture (MAPA), The Federal University of Western Pará (UFOPA), Chico Mendes Institute for Biodiversity Conservation (ICMBIO), Hope Foundation (IESPES), EMBRAPA Eastern Amazon, Tapajós Community Leaders, The Nature Conservancy (TNC) and Luiz de Quieroz College of Agriculture (ESALQ-USP) |



**Table 2: Structural metrics of Fuzzy Cognitive Maps analysed.**

| Structural Metric | Definition | Source |
|---|---|---|
| Outdegree *(od(vi))* | Cumulative total of transmitted connection weights from each component (horizontal cumulative sum within adjacency matrix). | Wasserman and Faust 1994 |
| Indegree *(id(vi))* | Cumulative total of received connection weights to each component (vertical sum within adjacency matrix). | Wasserman and Faust 1994 |
| Receiver variables *(R)* | Components that receive connections from other components but does not influence others through outward connections (components with zero *od(vi)*) | Özesmi and Özesmi 2003 |
| Transmitter variables or drivers *(T)* | Components that solely influences other components through outward connections but does not receive connections (components with zero *id(vi)*) | Özesmi and Özesmi 2003 |
| Ordinary variables *(O)* | Components that both influence and are influenced upon within the system | Özesmi and Özesmi 2003 |
| Density *(D)* | Number of connections ($C$) divided by the maximum number of possible connections between a number N of components $$D = \frac{C}{N(N-1)}$$ | Devisscher et al. 2016; Hage and Harary, 1983 |
| Complexity *(CM)* | Number of receiver components *(R)* divided by the number of transmitters *(T)*. A receiver being a $$CM = \frac{R}{T}$$ | Devisscher et al. 2016; Özesmi and Özesmi 2004 |
| Betweeness *(Bw)* | Betweenness is a centrality measure of influence of a node within a network. This measure quantifies the number of times a node acts as an intermediary along the shortest path between two other nodes. | Freeman, 1977; Brandes, 2001 |
| Page Rank *(PR)* | Used to determine a node's relevance or importance. PageRank value for a node $u$ is dependent on the PageRank values for each node v contained in the set $Bu$ (the set containing all nodes linking to node $u$), divided by the number L($v$) of links from page $v$. $$PR\,(u) = \sum_{v \in B_u} \frac{PR(v)}{L\,(v)}$$ | Page et al., 1999; Berkhim, 2005; this study |





5   **Table 3: Overview of the simulated scenarios.**

| Scenario | Description | Case Study | | | | | |
|---|---|---|---|---|---|---|---|
| | | Guarayos | | | Tapajós | | |
| | | Component | Value change (with respect to steady state baseline) | Scenario fixed value | Component | Value change (with respect to steady state baseline) | Scenario fixed value |
| 1. Governance Reform | Introduces institutional and governance improvements to the system, with policies widely implemented and governmental communication and efficiency improved | Lack of understanding, application and coordination of laws | Decrease | 0.4 | Lack of governmental co-ordination | Decrease | 0.4 |
| | | Poor administration by community leaders | Decrease | 0.3 | Lack of efficiency in policies for subsistence Farming | Decrease | 0.4 |
| | | | | | Lack of public policy | Decrease | 0.4 |
| 2. Techno-Social Reform | Considers a system in which investments are made in technical and social capital through capacity building, improvements in education and protection of traditional communities. | Lack of awareness of environmental problems | Decrease | 0.2 | Lack of environmental awareness | Decrease | 0.2 |
| | | Land encroachment | Decrease | 0.3 | Lack of technical training and assistance | Decrease | 0.3 |
| | | Loss of subsistence agriculture by Guarayos Communities | Decrease | 0.3 | Technical and productive capacity | Increase | 0.8 |
| | | | | | Lack of protection of traditional communities | Decrease | 0.3 |
| 3. Conservation | Focusses solely upon conserving the environment, with no consideration of socio-economic development. | Compliance with land zoning | Increase | 0.8 | Environmental monitoring | Increase | 0.8 |
| | | Application of forest law | Increase | 0.8 | | | |
| 4. Agricultural Development | Encourages extractionist activities, such as agricultural expansion, encouraged to improve the socio-economic conditions of each region. | Agricultural expansion | Increase | 0.9 | Agricultural expansion | Increase | 0.9 |
| | | Application of agricultural Law | Increase | 0.8 | Use of agrochemicals | Increase | 0.9 |
| | | Agricultural intensification | Increase | 0.8 | | | |
| 5. Climate Change | The current system where climate change continues. | Climate change | Increase | 1.0 | Climate change | Increase | 1.0 |



**Table 4: Guarayos and Tapajós fuzzy cognitive maps indices. Standard deviations shown in brackets and maximum values of the centrality indices.**

| Indices | Guarayos | Tapajós |
|---|---|---|
| Components | 29 | 32 |
| Transmitters | 7 | 9 |
| Receivers | 4 | 3 |
| Ordinary | 18 | 20 |
| Connections | 44 | 50 |
| Average Connection Weight (SD) | 0.57 (0.26) | 0.61 (0.22) |
| Connections per Component | 1.52 | 1.56 |
| Density | 0.052 | 0.048 |
| Complexity | 0.57 | 0.33 |
| Betweeness | 0.21 | 0.09 |
| PageRank | 0.13 | 0.17 |



**List of figures**

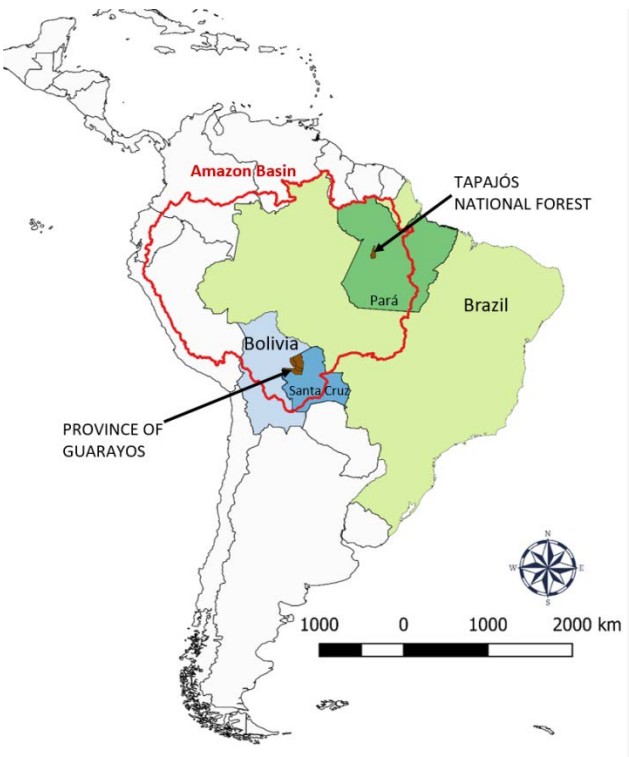

5    **Figure 1: Location of the case study sites (the Province of Guarayos in Bolivia and the Tapajós National Forest in Brazil). Case studies shaded in brown. The Department of Santa Cruz (Bolivia) shaded in dark blue and the State of Pará (Brazil) in dark green. The extent of the Amazon Basin is outlined in red.**



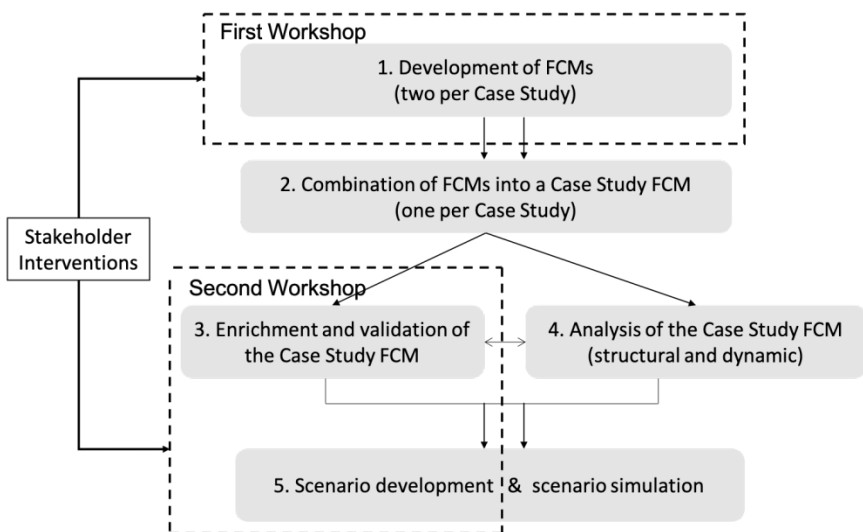

**Figure 2: Methodological steps in the research.**



**Figure 3: Frequencies of Betweeness (A) and PageRank values (B) in both case studies: Guarayos (red) and Tapajós (green)**





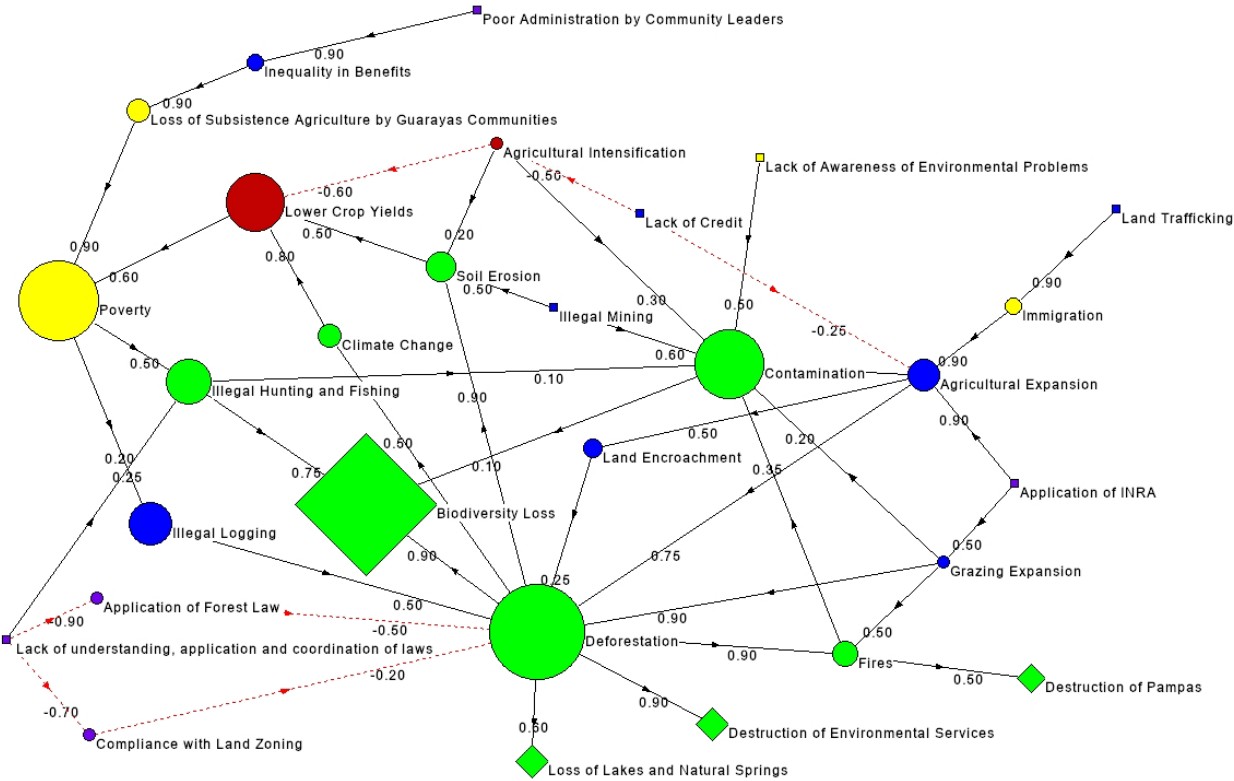

**Figure 4: Network visualization of the Case Study FCM developed by stakeholders in Guarayos. Size of each component represents their page rank. Solid black lines represent positive connection weights and red dotted lines negative. Shape of each component represents its type (square=transmitter, circle=ordinary and diamond=receiver) and colours their grouping (green=environmental, blue=economic, yellow=social, purple=political/ institutional and red=technical).**





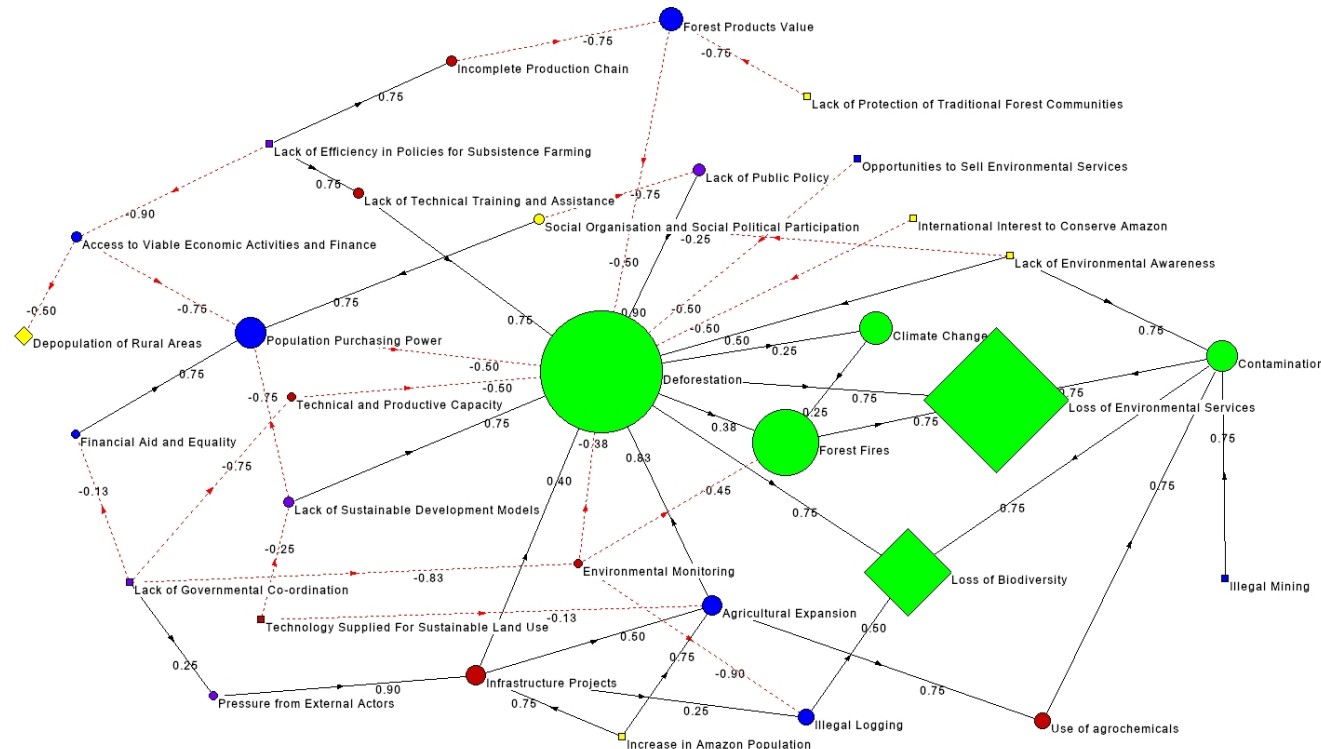

**Figure 5**: **Network visualization of the Case Study FCM developed by stakeholders in Tapajós. Size of each component represents their page rank. Solid black lines represent positive connection weights and red dotted lines negative. Shape of each component represents its type (square=transmitter, circle=ordinary and diamond=receiver) and colours their grouping (green=environmental, blue=economic, yellow=social, purple=political/ institutional and red=technical).**





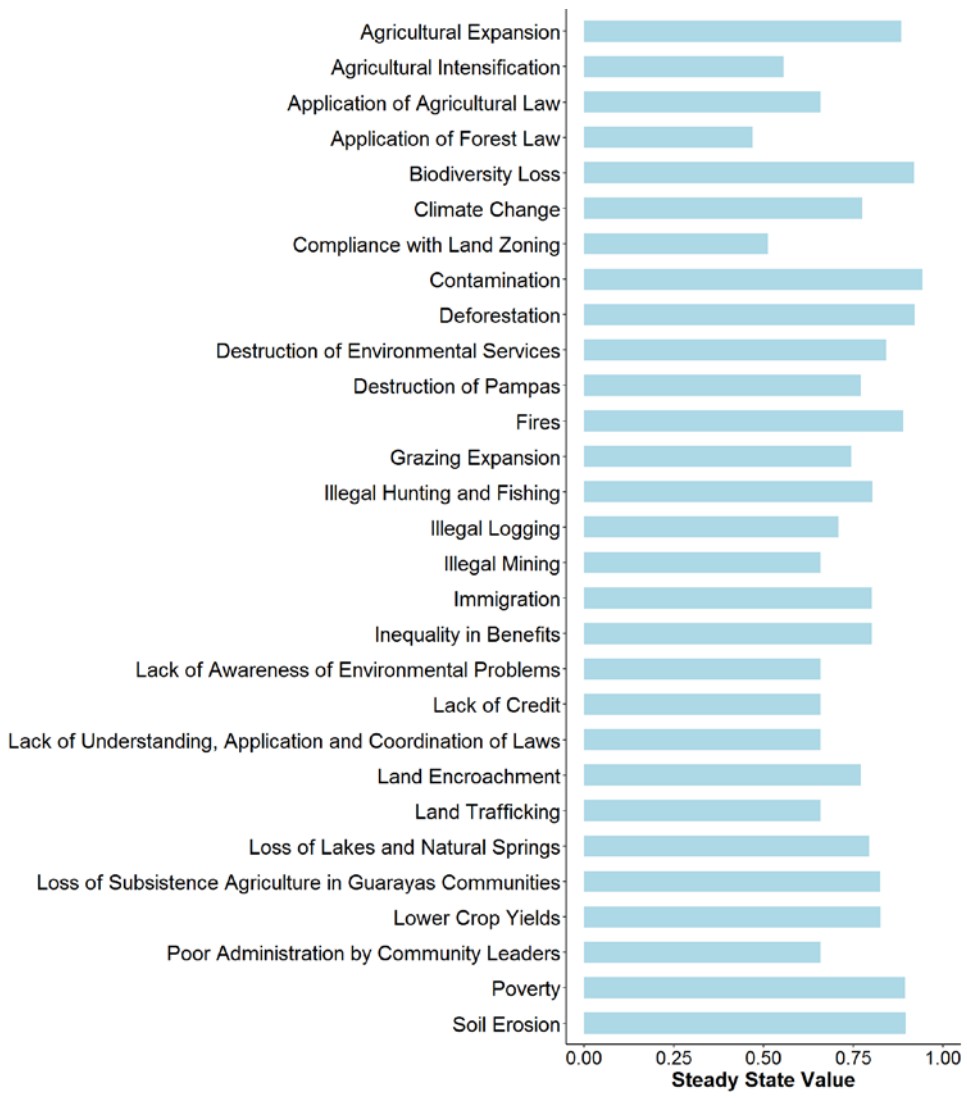

**Figure 6: Component values for Guarayos Case Study FCM under steady state 'baseline' conditions. Values close to 0 represent a strong decrease in the component, whilst values closer to 1 represent a strong increase.**



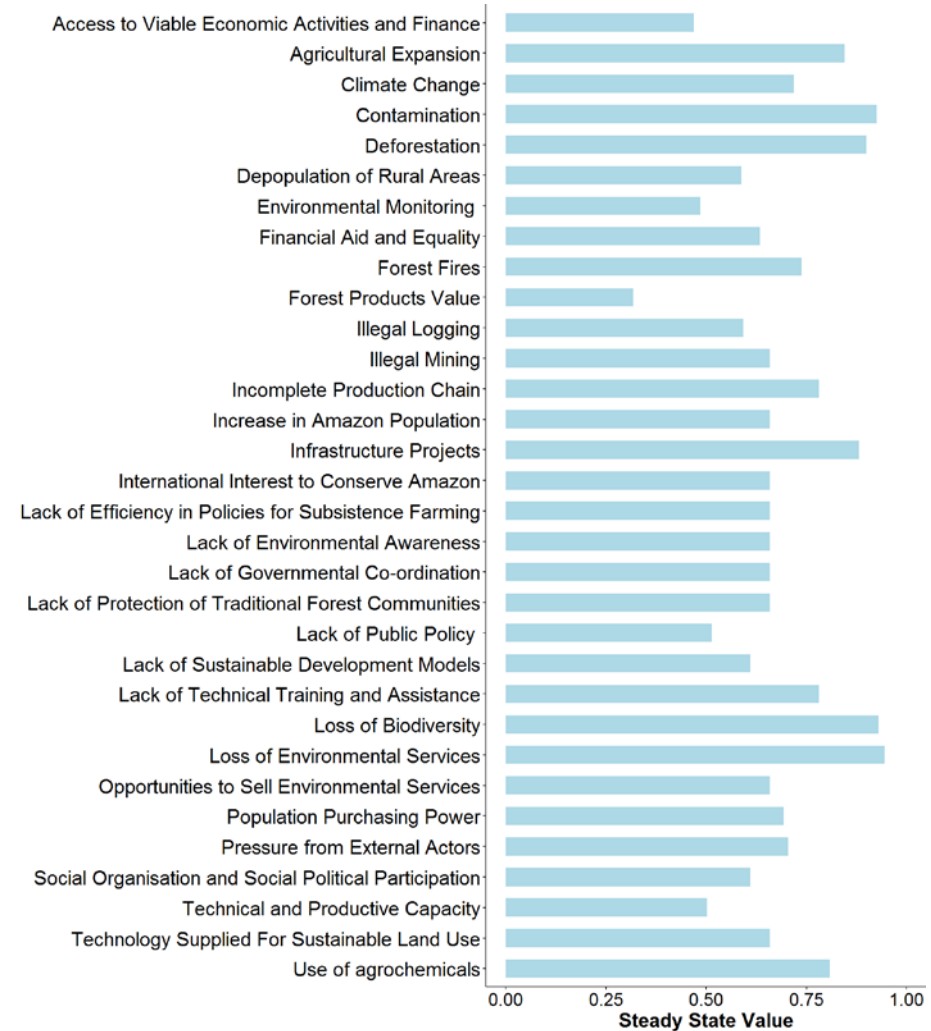

**Figure 7: Component values for Tapajós Case Study FCM under steady state 'baseline' conditions. Values close to 0 represent a strong decrease in the component, whilst values closer to 1 represent a strong increase.**





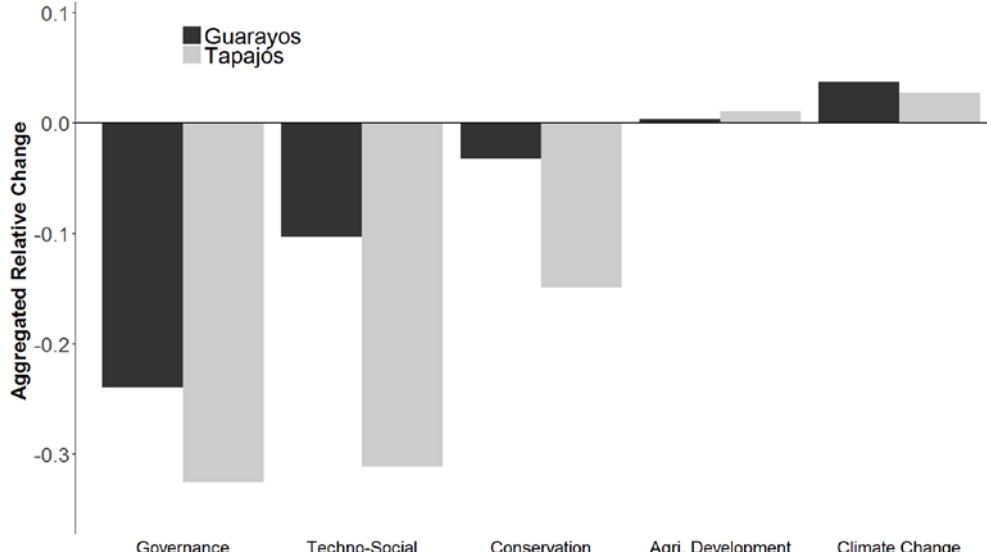

**Figure 8: Aggregated relative change and response of scenarios, compared with baseline. Negative values represent a 'desirable' change in the system. Positive values represent an 'undesirable' change in the system.**





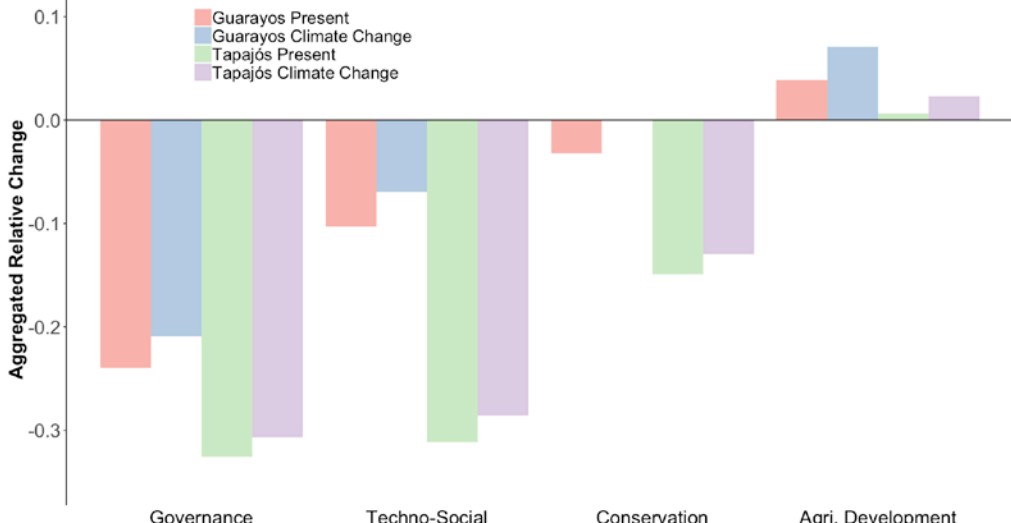

**Figure 9: Aggregated relative change and response of scenarios under present climatic conditions, and climate change. Negative values represent a 'desirable' change in the system. Positive values represent an 'undesirable' change in the system.**

