# Peer review of "Examining the sustainability and development challenge in agricultural-forest frontiers of the Amazon Basin through the eyes of locals"

_Natural Hazards and Earth System Sciences, 2019_

## Referee Comment (RC1) · Anonymous Referee #1 · 3 Jun 2019

**General comments:**

The paper presents a novel methodology for the analysis of interactions between the socio-economic and environmental aspects of a region. It is tested in two different regions with similar problems linked to deforestation. The paper addresses relevant scientific questions within the scope of NHESS, presenting novel concepts and tools, which are usable in other contexts in the world. The methods used are clearly explained and the results support the interpretation and conclusions of the paper. The description of the data, the methods used, the calculations made and the results obtained are

sufficiently complete and accurate to allow their reproduction. The title clearly and unambiguously reflects the contents of the paper, while the abstract provides a concise, complete and unambiguous summary of the work done and the results obtained. The overall presentation is well structured, clear and easy to understand by a wide and general audience. The paper is, as a whole, of a high quality, although some aspects could still be improved.

\*\*Specific comments:

Regarding the structure of the paper, section 2.1 Description of the study area should be part of the introduction, not of the methodology section.

The paper is very well written, with detailed explanations of the method and the results. However, it would improve readability if some parts were shortened. The introduction and the study area description, for example, are too long and contains irrelevant information that could be deleted, such as mean annual temperatures or precipitation, which are not needed and it is sufficient to know the type of climate for the purpose of the article.

Some additional information on the scenario selection should be included. In section 2.4, it should be explained why are those scenarios selected and how are they translated into the models? In particular, the 'climate change' scenario is too simplistic and it should not be presented as a scenario itself but only as an important element to analyse together with the development scenarios (as it is mentioned in page 9, lines 4-5).

Page 6, line 23: the authors mention two focus groups per study area without justifying why. Please briefly clarify why 2 focus groups were organised instead of one, which could have avoided the merging phase. It is also not clarified if the 2 focus groups were similarly composed, in terms of stakeholder groups.

Page 7, lines 11-15: it is not clear what the 'centrality' concept is; please add a short

clarification.

Page 8, line 1: it would be easier to understand the equation elements with a very small figure containing the components (ci, cj) with the edges and the weights in a visual way.

Page 8, line 26: how are values between 0-1 determined?

Page 10, lines 15-24: The paragraph is presented as facts, but this is the perceived view of stakeholders and it does not mean it is a demonstrated truth. Please rephrase so that it is clear that authors are presenting the reality perceived by stakeholders.

Table 1: the list of stakeholder groups is long and not easy to understand by outsiders. It would be easier for the reader if the table added a column (or some other feature) classifying them by wider types of stakeholder groups, such as 'farmers, environmentalists, local government. . ..'.

Table 3: I would remove the climate change scenario, as explained in previous comments

**Technical corrections:

Page 5, line 32, add "concept" after "FCM". Page 6, line 2, add "called" before "nodes". Page 6, line 3, add "The weight of" before "these relationships". Page 6, line 13, replace "scare" by "scarce". Page 9, line 32: remove 'and problems', it is redundant. Page 11, line 7: remove 'them' after 'studies'. Page 13, line 9: introduce 'situation of' (or something similar) between 'worsen' and 'region'. Page 13, line 25: move 'absent or ineffective' before 'social and governance'. Page 13, line 27: replace 'are' by 'is' after 'deforestation'. Table 2: font size is too small for reading Figures 4, 5: font size of the maps' elements is too small

---

## Author Comment (AC1) · 10 Jul 2019

We thank Reviewer 1 for the many insightful comments and suggestions. Please see the supplement.zip file in which both the responses to the Reviewer 1 and the revised manuscript (nhess-2019-144) are included.

Please also note the supplement to this comment:
https://www.nat-hazards-earth-syst-sci-discuss.net/nhess-2019-144/nhess-2019-144-AC1-supplement.zip

---

## Referee Comment (RC2) · Pei-Lin Yu (Referee) · 12 Sep 2019

Peer Review Comments for Blanco-Guitierrez et al. 2019

Journal: Natural Hazards Earth Systems Science; Special Issue: Remote sensing, modelling-based hazard and risk assessment, and management of agro-forested ecosystems

Title: Examining the sustainability and development challenge in agricultural-forest frontiers of the Amazon Basin through the eyes of locals Author(s): Irene Blanco-

Gutiérrez et al.

COMMENTS

1. Initial paragraph or section evaluating the overall quality of the discussion paper ("general comments").

Scientific Significance: The manuscript represents a substantial contribution to the understanding of natural hazards and their consequences through the use of the network analysis called Fuzzy Cognitive Mapping.

Scientific Quality: The scientific and/or technical approaches and the applied methods are largely valid in my judgment, although I am not a network analyst. The authors seem to 'lump' two very different communities together for comparative purposes; would benefit from describing the cultural/ethnic/socioeconomic makeup of the focus groups sampled. In addition, the importance of linguistic and cultural variability in understanding of terms such as climate change should be clearly addressed.

Presentation Quality: The scientific data, results and conclusions are presented in a clear, concise, and well-structured way.

2. Individual scientific questions/issues ("specific comments").

P. 2 Lines 10-15. Consider updating this introduction with urgency of environmental degradation such as the recent megafires in Amazon.

P. 4 line 4, worth mentioning that Brazilian governance structure demonstrates the volatility of politics in Amazonian countries, and the relative disengagement of the larger global community. This could also be discussed briefly on p. 14, line 15-16.

P. 5 Line 10. What is the cultural background/ethnicity of these ribeirinhos? Seems that these communities in Bolivia and Brazil would likely have some important differences.

P. 12 Lines 10-15. With regard to climate change it's possible that there are cultural, linguistic, inter-group, or even individual differences in perceptions of the meaning of

the term 'climate change'. Please address this.

P. 13-14. In discussion mentions unanticipated results for climate change which reinforces my comment above. In my experience conducting climate change oriented interviews with indigenous gardeners of the sub-tropics, interviewees stated clearly that climate change is not relevant because 'the weather is always changing'. Thus it's worth asking if concepts of climate change amongst Western scientists might not apply to traditional communities.

3. Compact listing of purely technical corrections at the very end ("technical corrections": typing errors, etc.). Not included.

Table 3 has misspelling 'focusses'

P. 14 Section heading: I think it should read "Effecting Change..."
* * *

---

## Referee Comment (RC3) · Anonymous Referee #3 · 2 Oct 2019

General Comments:

1. The paper presents two interesting case studies from Amazon countries, where the FCM approach was adopted to understand the perceptions of local actors about their environmental context. As result, different networks and scenarios were present to debate how local actor from each region could reacts to the sustainability and development challenges.

2. In the introduction section, the narrative conducts the reader to the importance of

two groups of stakeholders in Bolivian (Guarayos indigenous communities) and Brazilian Amazon (Tapajós riverine communities). An important point in this kind of modeling approach is the choice of stakeholders to represent the multiplicity of actors and perceptions for tackling the problem analyzed. Considering this, some question come up:

- Do the authors think that the riverine and indigenous communities were well represented in the groups of stakeholders that participated in the workshop?

- Do the cognitive maps represent the vision of these groups?

3. The description of the study area is long. The authors could be more focused on providing elements to support the research questions and the results (especially, the scenarios). For example, the social, cultural and political contexts experienced by stakeholders that can influence the networks structures or different responses to scenarios.

4. Regarding the description of the study area, details of temperature, precipitation and vegetation are not relevant in this section, unless they are used in the design of climate change scenarios (that would be interesting).

5. The section 2 should focus a little more on describing the workshops. Given that the stakeholder participants within each case study seem to be diverse and present even contrasting view on development and conservation, some issues need to be clarified, such as:

- How the authors selected the stakeholders groups? How conducted the process of identifying the components to be included in the model? How do the participants identify the degree of influence between components (high, medium, etc.)? What were the most important components mentioned in the workshops?

- It is unclear how component values were obtained during the workshops (were individual or group responses?). How have the authors converted the cognitive maps in the adjacent matrix? I mean, how the strength of the interactions among components (the

weighted values) was defined? How the contrasting view of the problem was converted in a single value of influence? I suggest the authors to provide the ranges of model parameters/variables presented during the workshops to show contrasting views.

6. Scenario section (2.4) is not clear. The authors could provide more details about the scenario conception and the stakeholders' contribution.

- How were climate change identified by stakeholders (changes in temperature, extreme climate events, precipitation, river level, floods, forest fires, soil erosion, etc.) and how were they translated it to the model? The climate change scenarios are the same for the two study sites?

What climate changes were considered to be impacted by deforestation?

- Conservation strategies were resumed in one strategy in the Tapajós case study (Environmental Monitoring). It is not clear if the scenario components were defined in the workshop by the stakeholders or by the authors. Anyway, I see as a problem reducing conservation strategies in a unique and passive action of monitoring. By doing this, conservation strategies may seem to have low impact to achieve desired changes, in comparison with the governance and techno-social reform.

7. The authors show in figure 2 that FCM was validated in the second workshop. How the validation procedure was carried out? Can the same participants in the first workshop validate the FCM they created themselves?

8. In the Dynamic analysis of FCM (3.2), some interesting results could be presented in respect to the model dynamics during the calculation to achieve the baseline situation. Does the system's identity remain the same after steady state analyses is conducted?

- Do the authors think that there is a relation between FCM complexity and the diversity of indigenous and riverine communities in the group of stakeholders?

Specific comments:

Page 2: Include more recent citations in introduction.

Page 5, Line 6: 'dense moist and wet forest types'. I suggest you include a classification system for the Amazon vegetation to describe the forest types.

Page 5, lines 7-9: I suggest you mention the decree n° 73.684, February 19 of 1974.

Page 5, Line 10: Cite data source (reference) in respect to the number of 'ribeirinhos' and 16 communities mostly along Tapajós river.

Page 6, Lines 20 – 25: How were focus groups defined?

Page 7: Last paragraph: I suggest to present the adjacent matrix in the supplementary material.

Page 9: How much components were included in the model by the workshop participants?

---

## Author Comment (AC2) · 8 Nov 2019

We thank Reviewer 2 for the many insightful comments and suggestions. Please see the supplement.zip file in which the responses to the Reviewer 2, the revised manuscript (nhess-2019-144) and supplementary material are included.

Please also note the supplement to this comment: https://www.nat-hazards-earth-syst-sci-discuss.net/nhess-2019-144/nhess-2019-144-AC2-supplement.zip

---

## Author Comment (AC3) · 8 Nov 2019

We thank Reviewer 3 for the many insightful comments and suggestions. Please see the supplement.zip file in which the responses to the Reviewer 3, the revised manuscript (nhess-2019-144) and the supplementary material are included.

Please also note the supplement to this comment: https://www.nat-hazards-earth-syst-sci-discuss.net/nhess-2019-144/nhess-2019-144-AC3-supplement.zip

---

## Author Response (AR1)

This document summarizes all changes made in the manuscript:

1- Affiliation of the author 'Marisol Toledo' changed to:

   Museo de Historia Natural Noel Kempff Mercado - Universidad Autónoma Gabriel René Moreno (UAGRM), Av. Irala 565, casilla 2489, Santa Cruz de la Sierra, Bolivia , Santa Cruz de la Sierra, Bolivia

2- Supplementary material

   According to Referee #3 and the Editor, we have included the 2 adjacency matrix (S1 Guarayos and S2 Tapajos) in the Supplementary Material.

3- Anonymous Referee #1

Authors' responses to review comments are in ***red, bold, italics.***

- Changes made and submitted by the authors: 10 July 2019
References to lines and pages have been updated (4 December 2019) to match the final version of the text (it includes changes made according also to Referee #2 and #3).

**General comments:

The paper presents a novel methodology for the analysis of interactions between the socio-economic and environmental aspects of a region. It is tested in two different regions with similar problems linked to deforestation. The paper addresses relevant scientific questions within the scope of NHESS, presenting novel concepts and tools, which are usable in other contexts in the world. The methods used are clearly explained and the results support the interpretation and conclusions of the paper. The description of the data, the methods used, the calculations made and the results obtained are sufficiently complete and accurate to allow their reproduction. The title clearly and unambiguously reflects the contents of the paper, while the abstract provides a concise, complete and unambiguous summary of the work done and the results obtained. The overall presentation is well structured, clear and easy to understand by a wide and general audience. The paper is, as a whole, of a high quality, although some aspects could still be improved.

***Thank you very much for the review and positive feedback.***

**Specific comments:

Regarding the structure of the paper, section 2.1 Description of the study area should be part of the introduction, not of the methodology section.

***Thank you for this comment. We agree that the description of the study area is not really part of the methodology. We have added a new point '2 Description of the study area' that goes after '1 Introduction' and before '3 Methodology'. Authors think that it is better not to merge Sections '1 Introduction' and '2 Description of the study area' because they present self-contained information that goes from the general to the specific.***

The paper is very well written, with detailed explanations of the method and the results. However, it would improve readability if some parts were shortened. The introduction and the study area description, for example, are too long and contains irrelevant information that could be deleted, such as mean annual temperatures or precipitation, which are not needed and it is sufficient to know the type of climate for the purpose of the article.

*Good suggestion. We have deleted irrelevant information and shortened the introduction (from 58 to 51 lines) and the description of the study area (from 64 to 38 lines) to improve readability. See sections 1 and 2 of the revised manuscript.*

Some additional information on the scenario selection should be included. In section 2.4, it should be explained why are those scenarios selected and how are they translated into the models? In particular, the 'climate change' scenario is too simplistic and it should not be presented as a scenario itself but only as an important element to analyse together with the development scenarios (as it is mentioned in page 9, lines 4-5).

*Thank you for the observations. We have now described the selection process used for the scenarios (Pg 7 line 29- Pg 8 lines 1-6). Following your suggestion we have now provided a more extended explanation of the translation of these scenarios into the model (Pg 7: lines 7-12). Further, we have also removed the climate change scenario as being an independent scenario and just defined it as an additional element for the scenarios.*

Page 6, line 23: the authors mention two focus groups per study area without justifying why. Please briefly clarify why 2 focus groups were organised instead of one, which could have avoided the merging phase. It is also not clarified if the 2 focus groups were similarly composed, in terms of stakeholder groups.

*Agreed. We have clarified the organization (Pg 5: lines 26-28) and composition of the focus groups (Pg 5: lines 28-31).*

Page 7, lines 11-15: it is not clear what the 'centrality' concept is; please add a short clarification.

*Agreed. Done (Pg 6: lines 26-28).*

Page 8, line 1: it would be easier to understand the equation elements with a very small figure containing the components ($c_i$, $c_j$) with the edges and the weights in a visual way.

*Although we agree that inclusion of this information ($c_i$, $c_j$) may provide greater support to the work, however an adequate explanation of such information would require an extension section and we believe would probably be more confusing than aiding. The values for the edges and weights of the components are included in Figures 4 and 5.*

for Page 8, line 26: how are values between 0-1 determined?

*We have now better explained how these values were determined: "Following Reckien (2014), we translated each scenario into the analysis through the manipulation of individual component state vector values (A of Eq. 1: Sect. 2.3.2). (Table 3). For each scenario, we identified components which we assumed would be directly affected by the scenario implementation. For these selected components, their values were fixed between 0-1, depending upon the scale of the scenario's impact. If we assumed a strong increase in the selected component, its state vector value was set to 1, whilst a strong decrease was set to 0. Intermediate values represent less intense increases or decreases. All other components had their values set to 0."*

Page 10, lines 15-24: The paragraph is presented as facts, but this is the perceived view of stakeholders and it does not mean it is a demonstrated truth. Please rephrase so that it is clear that authors are presenting the reality perceived by stakeholders.

*Agreed. Done (Pg 9: lines 26-33 and Pg 10 lines 1-2).*

Table 1: the list of stakeholder groups is long and not easy to understand by outsiders. It would be easier for the reader if the table added a column (or some other feature) classifying them by wider types of stakeholder groups, such as 'farmers, environmentalists, local government: : :.'.

*Very useful comment. Done. In Table 1, a column has been added with the stakeholder group (policy/administration, private sector, non-governmental organization, research)*

Table 3: I would remove the climate change scenario, as explained in previous comments

*Agreed. Done.*

**Technical corrections:

Page 5, line 32, add "concept" after "FCM".
*Thank you. Done. Pg 5: line 3*

Page 6, line 2, add "called" before "nodes".
*Thank you. Done. Pg 5: line 6*

Page 6, line 3, add "The weight of" before "these relationships".
*Thank you. Done. Pg 5: line 8*

Page 6, line 13, replace "scare" by "scarce".
*Thank you. Done. Pg 5: line 17*

Page 9, line 32: remove 'and problems', it is redundant.
*Thank you. Done. Pg 9: line 11*

Page 11, line 7: remove 'them' after 'studies'.
*Thank you. Done. Pg 10: line 18*

Page 13, line 9: introduce 'situation of' (or something similar) between 'worsen' and 'region'.
*Thank you. Done. Pg 12: line 18*

Page 13, line 25: move 'absent or ineffective' before 'social and governance'.
*Changed by 'weak social and governance support structures'. Pg 12: line 34*

Page 13, line 27: replace 'are' by 'is after 'deforestation'.
*Thank you. Done. Pg 13: line 2*

Table 2: font size is too small for reading
*Agreed. Font sized has been increased*

Figures 4, 5: font size of the maps' elements is too small
*Agreed. Font sized has been increased*

4- Anonymous Referee #2 Pei-Lin Yu

- Changes made and submitted by the authors: 4 November 2019

1. Initial paragraph or section evaluating the overall quality of the discussion paper ("general comments").
Scientific Significance: The manuscript represents a substantial contribution to the understanding of natural hazards and their consequences through the use of the network analysis called Fuzzy Cognitive Mapping.
Scientific Quality: The scientific and/or technical approaches and the applied methods are largely valid in my judgment, although I am not a network analyst. The authors seem to 'lump' two very different communities together for comparative purposes; would benefit from describing the cultural/ethnic/socioeconomic makeup of the focus groups sampled. In addition, the importance of linguistic and cultural variability in understanding of terms such as climate change should be clearly addressed.
Presentation Quality: The scientific data, results and conclusions are presented in a clear, concise, and well-structured way.

*Authors sincerely thank the referee for the review, constructive comments and positive feedback. Suggested improvements are much appreciated and they have been addressed below.*

2. Individual scientific questions/issues ("specific comments").

P. 2 Lines 10-15. Consider updating this introduction with urgency of environmental degradation such as the recent megafires in Amazon.
*Good suggestion. Agreed and included (Pg. 2: lines 6-7)*

P. 4 line 4, worth mentioning that Brazilian governance structure demonstrates the volatility of politics in Amazonian countries, and the relative disengagement of the larger global community. This could also be discussed briefly on p. 14, line 15-16.

*Thank you. We have added a sentence about it in the introduction (Pg. 2: lines 5-6) because we think that it is in fact very important, but applicable to all the Amazonian countries, not only Brazil. In addition, we believe this has already been touched upon within the discussion (Pg 14, line 35-Pg 15 line 2).*

P. 5 Line 10. What is the cultural background/ethnicity of these ribeirinhos? Seems that these communities in Bolivia and Brazil would likely have some important differences.

*Thanks for highlighting this. We were instructed to reduce the content concerning the communities by a previous reviewer. However, as suggested, we have clarified the background of the 'ribeirinhos' in Pg. 4, lines 16-17, in Section 2 'Description of the study area', and included relevant information in Pg. 4: lines 10-12 and line 24 .*

*The studied communities (Guarayos in Bolivia and Ribeirinhos in Brazil) are culturally different, but their conditions are largely similar. Both live at the edge of the agricultural frontier, are reliant upon natural resources for incomes, face high levels of poverty, and are increasingly threatened by outside forces. The differences perceptions of the present situation in Guarayos and in Tapajos are already included in the different FCMs developed during the workhops (Figures 4 and 5).*

P. 12 Lines 10-15. With regard to climate change it's possible that there are cultural, linguistic, inter-group, or even individual differences in perceptions of the meaning of the term 'climate change'. Please address this.

*Agreed. We have included this caveat in the text. "This finding may also reflect the distinct cultural and linguistic meaning or representations of climate changes (e.g drought, flooding) across the two sites." (Pg 13: lines 9-10).*
*However, we should point out that the FCMs are group maps and therefore 'agreed' or 'consensual' maps developed during the workshops. Discussions between stakeholders were carefully guided by a facilitator, who helped to reach consensus. These types of exercises are not meant to identify (individual) contrasting views, to do so it is better to develop individual FCMs or other methodologies. Also, as part of the FCM methodology (Pg 5) a number of components considered to be representing similar features were merged. Therefore, components like reduced rains or increased droughts are included under this catch-all phrase of climate change. We agree that the need for highlighting linguistic and cultural distinctions is definitely relevant, but we don't believe it will have greatly affected the results here.*

P. 13-14. In discussion mentions unanticipated results for climate change which reinforces my comment above. In my experience conducting climate change oriented interviews with indigenous gardeners of the sub-tropics, interviewees stated clearly that climate change is not relevant because 'the weather is always changing'. Thus it's worth asking if concepts of climate change amongst Western scientists might not apply to traditional communities.

*Interesting point. We completely agree that cultural perspectives will have a considerable impact on perceptions of concepts like climate change. However, it does not apply to our study. As is common with the Fuzzy Cognitive Mapping method, similar components are grouped together. The stakeholders in both Brazil and Bolivia mentioned an array or terms (e.g.*

*increasing drought, reduced rains, increasing floods, weather instability), but they decided during the workshops to use the word 'climate change' to catch all terms. Further, in follow up meetings (Varela-Ortega et al., 2014) stakeholders validated this combination as being accurate to the current situation.*

3. Compact listing of purely technical corrections at the very end ("technical corrections": typing errors, etc.). Not included.

Table 3 has misspelling 'focusses'
*The manuscript has been written in British English, making this spelling appropriate.*

P. 14 Section heading: I think it should read "Effecting Change..."
*Thank you, we have change the word 'affecting' by 'encouraging' . Section heading 5.2 in Pg. 13.*

5- Anonymous Referee #3

- Changes made and submitted by the authors: 4 November 2019

*We thank Reviewer 3 for the many insightful comments and suggestions.*

General Comments:

1. The paper presents two interesting case studies from Amazon countries, where the FCM approach was adopted to understand the perceptions of local actors about their environmental context. As result, different networks and scenarios were present to debate how local actor from each region could reacts to the sustainability and development challenges.

2. In the introduction section, the narrative conducts the reader to the importance of two groups of stakeholders in Bolivian (Guarayos indigenous communities) and Brazilian Amazon (Tapajós riverine communities). An important point in this kind of modeling approach is the choice of stakeholders to represent the multiplicity of actors and perceptions for tackling the problem analyzed. Considering this, some questions come up:
- Do the authors think that the riverine and indigenous communities were well represented in the groups of stakeholders that participated in the workshop?
*Good point. Yes, riverine and indigenous communities were well represented in the workshops. See Table 1. In both cases (Guarayos in Bolivia and Tapajós in Brazil), key representatives of the indigenous communities (with the ability to make and to influence decisions) attended the workshops. E.g., in Guarayos (Bolivia), several representatives of the Organisation Centre of Guarayo Native People (COPNAG), which is the most powerful and influential indigenous association in the region attended the workshops. Similarly, in Tapajós (Brazil), the representative of all indigenous communities of the Flona (who lived in Communidade do Maguari) attended the workshop, together with other indigenous community heads. Indigenous communities were reached by the local teams of the ROBIN project (IBIF in Bolivia, and EMBRAPA in Brazil; researchers of both teams are co-authors*

*of the paper), which are great connoisseurs in the area and have long experience working with indigenous communities.*

- Do the cognitive maps represent the vision of these groups?

*Yes. The maps include the vision of these groups. In fact, in Brazil, the representative of all indigenous communities of the Flona presented the FCM obtained in the plenary*

3. The description of the study area is long. The authors could be more focused on providing elements to support the research questions and the results (especially, the scenarios). For example, the social, cultural and political contexts experienced by stakeholders that can influence the networks structures or different responses to scenarios.

*Agree, thank you. Following your suggestion (and other similar from other reviewers) we have deleted irrelevant information (e.g., mean annual temperatures or precipitation; we have been told that it is sufficient to know the type of climate for the purpose of the article) and focused this section on the description of the socio-economic, cultural, and political context. In addition, as suggested by one of the reviewers, the description of the case study has been now separated from the methodology section. We have added a new point '2 Description of the study area' that goes after '1 Introduction' and before '3 Methodology'.*

4. Regarding the description of the study area, details of temperature, precipitation and vegetation are not relevant in this section, unless they are used in the design of climate change scenarios (that would be interesting).

*We have deleted this information. See previous comment (point 3).*

5. The section 2 should focus a little more on describing the workshops. Given that the stakeholder participants within each case study seem to be diverse and present even contrasting view on development and conservation, some issues need to be clarified, such as:

*Thank you very much for noticing this. We have made some changes following your suggestions (see below)*

- How the authors selected the stakeholders groups?

*Agree. We have clarified this. See page 5, lines 25-31 and Table 1*

How conducted the process of identifying the components to be included in the model? How do the participants identify the degree of influence between components (high, medium, etc.)?

*Agree. This has been explained in more detail in the manuscript. See page 6, lines 1-10*

What were the most important components mentioned in the workshops?

*The most important components are those reflected in the FCMs, and particularly those with the highest page rank (see Figures 4 and 5).*

- It is unclear how component values were obtained during the workshops (were individual or group responses?).

*Agree. They were group responses. This is now specified on page 6, lines 1-2 'the FCM developed represented stakeholder group knowledge' (Ösezmi and Ösezmi, 2004), and on page 6 line 11, FCMs are 'group maps'.*

How have the authors converted the cognitive maps in the adjacent matrix? I mean, how the strength of the interactions among components (the weighted values) was defined?

*The strength of the interactions among components was defined by the stakeholders in the workshops as described on page 6, lines 1-10*

How the contrasting view of the problem was converted in a single value of influence? I suggest the authors to provide the ranges of model parameters/variables presented during the workshops to show contrasting views.

*The FCMs are group maps and therefore 'agreed' or 'consensual' maps developed during the workshops. Discussions between stakeholders were guided by a facilitator, who helped to reach consensus. These types of exercises are not meant to identify contrasting views, to do so it is better to develop individual FCMs or other methodologies. Furthermore, the objective of the paper was not to dig on individual/contrasting views, but to have a clear picture of the common vision of the present in two communities (Guarayos and Tapajos) from different countries (Bolivia and Brazil), living on the edge of the agricultural frontier and confronting similar problems.*

6. Scenario section (2.4) is not clear. The authors could provide more details about the scenario conception and the stakeholders' contribution.

*Agree. Thank you. We have made some changes following your suggestions (see below)*

- How were climate change identified by stakeholders (changes in temperature, extreme climate events, precipitation, river level, floods, forest fires, soil erosion, etc.) and how were they translated it to the model?

*As is common with the Fuzzy Cognitive Mapping method, similar components were grouped together. The stakeholders in both Brazil and Bolivia mentioned an array or terms (e.g. increasing drought, reduced rains, increasing floods, weather instability), but they decided during the workshops to use the word 'climate change' to catch all terms. Further, in follow up meetings (Varela-Ortega et al., 2014) stakeholders validated this combination as being accurate to the current situation.*

*Following the suggestion of one of the reviewers, we have removed the climate change scenario as being an independent scenario and just defined it as an additional element for the scenarios. Also, following your comment, we have now provided a more extended explanation of the translation of these scenarios into the model (Pg 8: lines 7-18).*

The climate change scenarios are the same for the two study sites?
*Yes*
What climate changes were considered to be impacted by deforestation?
*Increasing drought, reduced rains, increasing floods, weather instability*

- Conservation strategies were resumed in one strategy in the Tapajós case study (Environmental Monitoring). It is not clear if the scenario components were defined in the workshop by the stakeholders or by the authors. Anyway, I see as a problem reducing conservation strategies in a unique and passive action of monitoring. By doing this, conservation strategies may seem to have low impact to achieve desired changes, in comparison with the governance and techno-social reform.

*The scenarios were first proposed by the authors, based on literature review, and then further defined by the stakeholders taking into account the limited number of factors included in the FCMs. We agree that the conservation scenario in the Tapajós case study may seem to be too reductionist, but stakeholders identified improved monitoring as the key environmental aspect to achieve a successful conservationist future. Stakeholders think that many conservation policies have already been developed and put in place, but their effectiveness has been limited due to insufficient monitoring and enforcement. Also, stakeholders think that a lot more remains to be done for improving institutional and governance systems, to protect traditional communities, support technical training, etc. Many aspects could be improved in this regard that could have positive impacts in the region. This is why the governance and techno-social scenario include changes in several components and the conservation scenario only in one.*

7. The authors show in figure 2 that FCM was validated in the second workshop. How the validation procedure was carried out? Can the same participants in the first workshop validate the FCM they created themselves?

*The people who participated in the second workshop are not exactly the same as those who participated in the first workshop. We were very careful to count with the same group of stakeholder, but the key representatives varied in some cases (due to agenda issues or changes in governmental bodies). Thus, the FCMs were validated by the same groups of stakeholders, not exactly by the same participants.*

*In the second workshop, the validation was performed by showing the stakeholders the processed FCM, including the dynamic analysis, and discussing with them the results. In both cases, Guarayos in Bolivia and Tapajos in Brazil, the main components of the FCMs remained unchanged, but stakeholders decided to change (increase/decrease) the strength of some links among components (e.g., in Guarayos, stakeholders decided to increase the weight given to the links 'illegal mining → soil erosion'; 'illegal mining → contamination').*

8. In the Dynamic analysis of FCM (3.2), some interesting results could be presented in respect to the model dynamics during the calculation to achieve the baseline situation. Does the system's identity remain the same after steady state analyses is conducted?

*Yes, the systems' identity remains the same. The steady state analysis considers the current situation of all variables. It is used to measure how a variable is changing (increasing, reducing, or stable based upon the value) in the system and you can also compare across variables (i.e whether deforestation is increasing, whilst forest law implementation is reducing) within the system. However, the system remains the same as the weighting applied to each variable is identical; the 'identity' would only change with the application of the scenarios, where the current situation of the system is altered. The iterations (calculation) of the model dynamics are irrelevant, the final result is what it is important and it is shown in Figures 6 and 7.*

- Do the authors think that there is a relation between FCM complexity and the diversity of indigenous and riverine communities in the group of stakeholders?

*No, we think that there is not such a relation. Key representatives transmitted a common voice for the indigenous and riverine communities. This is quite frequent; they used to have a common voice to make theirselves heard.*

Specific comments:

Page 2: Include more recent citations in introduction.
*Agree. Done*

Page 5, Line 6: 'dense moist and wet forest types'. I suggest you include a classification system for the Amazon vegetation to describe the forest types.
*Agree, defined now as dense terra firme (upland) tropical moist forest*

Page 5, lines 7-9: I suggest you mention the decree n_ 73.684, February 19 of 1974.
*Agree, included*

Page 5, Line 10: Cite data source (reference) in respect to the number of 'ribeirinhos' and 16 communities mostly along Tapajós river.
*Following the suggestion of another reviewer, we have deleted the number of communities and further specified the ethnical background of this riberirinhos*

Page 6, Lines 20 – 25: How were focus groups defined?
*This has now been detailed in Pg. 5, lines 26-31.*

Page 7: Last paragraph: I suggest to present the adjacent matrix in the supplementary material.
*Fist response (November 2019)→ We have tried to include the adjacency matrix as Tables in the Supplementary material, but it has been impossible, they are too big (29 lines x 29 columns in Guarayos, 32 lines x 32 columns in Tapajós). They are illegible*
*According to the Editor's comments (December 2019) → We have included the 2 adjacency matrix (S1 Guarayos and S2 Tapajos) in the Supplementary Material.*

Page 9: How much components were included in the model by the workshop participants?
*This is specified in Table 4 (second line), in Guarayos 29, in Tapajós 32*

[revised manuscript text omitted]

---

## Author Response (AR3)

This document summarizes all changes made in the manuscript:

1- Affiliation of the author 'Marisol Toledo' changed to:

Museo de Historia Natural Noel Kempff Mercado - Universidad Autónoma Gabriel René Moreno (UAGRM), Av. Irala 565, casilla 2489, Santa Cruz de la Sierra, Bolivia , Santa Cruz de la Sierra, Bolivia

2- Supplementary material

According to Referee #3 and the Editor, we have included the 2 adjacency matrix (S1 Guarayos and S2 Tapajos) in the Supplementary Material.

3- Anonymous Referee #1

Authors' responses to review comments are in *red, bold, italics.*

- Changes made and submitted by the authors: 10 July 2019
References to lines and pages have been updated (4 December 2019) to match the final version of the text (it includes changes made according also to Referee #2 and #3).

**General comments:

The paper presents a novel methodology for the analysis of interactions between the socio-economic and environmental aspects of a region. It is tested in two different regions with similar problems linked to deforestation. The paper addresses relevant scientific questions within the scope of NHESS, presenting novel concepts and tools, which are usable in other contexts in the world. The methods used are clearly explained and the results support the interpretation and conclusions of the paper. The description of the data, the methods used, the calculations made and the results obtained are sufficiently complete and accurate to allow their reproduction. The title clearly and unambiguously reflects the contents of the paper, while the abstract provides a concise, complete and unambiguous summary of the work done and the results obtained. The overall presentation is well structured, clear and easy to understand by a wide and general audience. The paper is, as a whole, of a high quality, although some aspects could still be improved.

*Thank you very much for the review and positive feedback.*

**Specific comments:

Regarding the structure of the paper, section 2.1 Description of the study area should be part of the introduction, not of the methodology section.

*Thank you for this comment. We agree that the description of the study area is not really part of the methodology. We have added a new point '2 Description of the study area' that goes after '1 Introduction' and before '3 Methodology'. Authors think that it is better not to merge Sections '1 Introduction' and '2 Description of the study area' because they present self-contained information that goes from the general to the specific.*

The paper is very well written, with detailed explanations of the method and the results. However, it would improve readability if some parts were shortened. The introduction and the study area description, for example, are too long and contains irrelevant information that could be deleted, such as mean annual temperatures or precipitation, which are not needed and it is sufficient to know the type of climate for the purpose of the article.

*Good suggestion. We have deleted irrelevant information and shortened the introduction (from 58 to 51 lines) and the description of the study area (from 64 to 38 lines) to improve readability. See sections 1 and 2 of the revised manuscript.*

Some additional information on the scenario selection should be included. In section 2.4, it should be explained why are those scenarios selected and how are they translated into the models? In particular, the 'climate change' scenario is too simplistic and it should not be presented as a scenario itself but only as an important element to analyse together with the development scenarios (as it is mentioned in page 9, lines 4-5).

*Thank you for the observations. We have now described the selection process used for the scenarios (Pg 7 line 29- Pg 8 lines 1-6). Following your suggestion we have now provided a more extended explanation of the translation of these scenarios into the model (Pg 7: lines 7-12). Further, we have also removed the climate change scenario as being an independent scenario and just defined it as an additional element for the scenarios.*

Page 6, line 23: the authors mention two focus groups per study area without justifying why. Please briefly clarify why 2 focus groups were organised instead of one, which could have avoided the merging phase. It is also not clarified if the 2 focus groups were similarly composed, in terms of stakeholder groups.

*Agreed. We have clarified the organization (Pg 5: lines 26-28) and composition of the focus groups (Pg 5: lines 28-31).*

Page 7, lines 11-15: it is not clear what the 'centrality' concept is; please add a short clarification.

*Agreed. Done (Pg 6: lines 26-28).*

Page 8, line 1: it would be easier to understand the equation elements with a very small figure containing the components ($c_i$, $c_j$) with the edges and the weights in a visual way.

*Although we agree that inclusion of this information ($c_i$, $c_j$) may provide greater support to the work, however an adequate explanation of such information would require an extension section and we believe would probably be more confusing than aiding. The values for the edges and weights of the components are included in Figures 4 and 5.*

for Page 8, line 26: how are values between 0-1 determined?

*We have now better explained how these values were determined: "Following Reckien (2014), we translated each scenario into the analysis through the manipulation of individual component state vector values (A of Eq. 1: Sect. 2.3.2). (Table 3). For each scenario, we identified components which we assumed would be directly affected by the scenario implementation. For these selected components, their values were fixed between 0-1, depending upon the scale of the scenario's impact. If we assumed a strong increase in the selected component, its state vector value was set to 1, whilst a strong decrease was set to 0. Intermediate values represent less intense increases or decreases. All other components had their values set to 0."*

Page 10, lines 15-24: The paragraph is presented as facts, but this is the perceived view of stakeholders and it does not mean it is a demonstrated truth. Please rephrase so that it is clear that authors are presenting the reality perceived by stakeholders.

*Agreed. Done (Pg 9: lines 26-33 and Pg 10 lines 1-2).*

Table 1: the list of stakeholder groups is long and not easy to understand by outsiders. It would be easier for the reader if the table added a column (or some other feature) classifying them by wider types of stakeholder groups, such as 'farmers, environmentalists, local government: : :.'.

*Very useful comment. Done. In Table 1, a column has been added with the stakeholder group (policy/administration, private sector, non-governmental organization, research)*

Table 3: I would remove the climate change scenario, as explained in previous comments

*Agreed. Done.*

**Technical corrections:

Page 5, line 32, add "concept" after "FCM".
*Thank you. Done. Pg 5: line 3*

Page 6, line 2, add "called" before "nodes".
*Thank you. Done. Pg 5: line 6*

Page 6, line 3, add "The weight of" before "these relationships".
*Thank you. Done. Pg 5: line 8*

Page 6, line 13, replace "scare" by "scarce".
*Thank you. Done. Pg 5: line 17*

Page 9, line 32: remove 'and problems', it is redundant.
*Thank you. Done. Pg 9: line 11*

Page 11, line 7: remove 'them' after 'studies'.
*Thank you. Done. Pg 10: line 18*

Page 13, line 9: introduce 'situation of' (or something similar) between 'worsen' and 'region'.
*Thank you. Done. Pg 12: line 18*

Page 13, line 25: move 'absent or ineffective' before 'social and governance'.
*Changed by 'weak social and governance support structures'. Pg 12: line 34*

Page 13, line 27: replace 'are' by 'is after 'deforestation'.
*Thank you. Done. Pg 13: line 2*

Table 2: font size is too small for reading
*Agreed. Font sized has been increased*

Figures 4, 5: font size of the maps' elements is too small
*Agreed. Font sized has been increased*

4-  Anonymous Referee #2 Pei-Lin Yu

- Changes made and submitted by the authors: 4 November 2019

1. Initial paragraph or section evaluating the overall quality of the discussion paper ("general comments").
Scientific Significance: The manuscript represents a substantial contribution to the understanding of natural hazards and their consequences through the use of the network analysis called Fuzzy Cognitive Mapping.
Scientific Quality: The scientific and/or technical approaches and the applied methods are largely valid in my judgment, although I am not a network analyst. The authors seem to 'lump' two very different communities together for comparative purposes; would benefit from describing the cultural/ethnic/socioeconomic makeup of the focus groups sampled. In addition, the importance of linguistic and cultural variability in understanding of terms such as climate change should be clearly addressed.
Presentation Quality: The scientific data, results and conclusions are presented in a clear, concise, and well-structured way.

*Authors sincerely thank the referee for the review, constructive comments and positive feedback. Suggested improvements are much appreciated and they have been addressed below.*

2. Individual scientific questions/issues ("specific comments").

P. 2 Lines 10-15. Consider updating this introduction with urgency of environmental degradation such as the recent megafires in Amazon.
*Good suggestion. Agreed and included (Pg. 2: lines 6-7)*

P. 4 line 4, worth mentioning that Brazilian governance structure demonstrates the volatility of politics in Amazonian countries, and the relative disengagement of the larger global community. This could also be discussed briefly on p. 14, line 15-16.

*Thank you. We have added a sentence about it in the introduction (Pg. 2: lines 5-6) because we think that it is in fact very important, but applicable to all the Amazonian countries, not only Brazil.  In addition, we believe this has already been touched upon within the discussion (Pg 14, line 35-Pg 15 line 2).*

P. 5 Line 10. What is the cultural background/ethnicity of these ribeirinhos? Seems that these communities in Bolivia and Brazil would likely have some important differences.

*Thanks for highlighting this. We were instructed to reduce the content concerning the communities by a previous reviewer. However, as suggested, we have clarified the background of the 'ribeirinhos' in Pg. 4, lines 16-17, in Section 2 'Description of the study area', and included relevant information in Pg. 4: lines 10-12 and line 24 .*

*The studied communities (Guarayos in Bolivia and Ribeirinhos in Brazil) are culturally different, but their conditions are largely similar. Both live at the edge of the agricultural frontier, are reliant upon natural resources for incomes, face high levels of poverty, and are increasingly threatened by outside forces. The differences perceptions of the present situation in Guarayos and in Tapajos are already included in the different FCMs developed during the workhops (Figures 4 and 5).*

P. 12 Lines 10-15. With regard to climate change it's possible that there are cultural, linguistic, inter-group, or even individual differences in perceptions of the meaning of the term 'climate change'. Please address this.

*Agreed. We have included this caveat in the text. "This finding may also reflect the distinct cultural and linguistic meaning or representations of climate changes (e.g drought, flooding) across the two sites." (Pg 13: lines 9-10).*
*However, we should point out that the FCMs are group maps and therefore 'agreed' or 'consensual' maps developed during the workshops. Discussions between stakeholders were carefully guided by a facilitator, who helped to reach consensus. These types of exercises are not meant to identify (individual) contrasting views, to do so it is better to develop individual FCMs or other methodologies. Also, as part of the FCM methodology (Pg 5) a number of components considered to be representing similar features were merged. Therefore, components like reduced rains or increased droughts are included under this catch-all phrase of climate change. We agree that the need for highlighting linguistic and cultural distinctions is definitely relevant, but we don't believe it will have greatly affected the results here.*

P. 13-14. In discussion mentions unanticipated results for climate change which reinforces my comment above. In my experience conducting climate change oriented interviews with indigenous gardeners of the sub-tropics, interviewees stated clearly that climate change is not relevant because 'the weather is always changing'. Thus it's worth asking if concepts of climate change amongst Western scientists might not apply to traditional communities.

*Interesting point. We completely agree that cultural perspectives will have a considerable impact on perceptions of concepts like climate change. However, it does not apply to our study. As is common with the Fuzzy Cognitive Mapping method, similar components are grouped together. The stakeholders in both Brazil and Bolivia mentioned an array or terms (e.g.*

*increasing drought, reduced rains, increasing floods, weather instability), but they decided during the workshops to use the word 'climate change' to catch all terms. Further, in follow up meetings (Varela-Ortega et al., 2014) stakeholders validated this combination as being accurate to the current situation.*

3. Compact listing of purely technical corrections at the very end ("technical corrections": typing errors, etc.). Not included.

Table 3 has misspelling 'focusses'
*The manuscript has been written in British English, making this spelling appropriate.*

P. 14 Section heading: I think it should read "Effecting Change..."
*Thank you, we have change the word 'affecting' by 'encouraging' . Section heading 5.2 in Pg. 13.*

5- Anonymous Referee #3

- Changes made and submitted by the authors: 4 November 2019

*We thank Reviewer 3 for the many insightful comments and suggestions.*

General Comments:

1. The paper presents two interesting case studies from Amazon countries, where the FCM approach was adopted to understand the perceptions of local actors about their environmental context. As result, different networks and scenarios were present to debate how local actor from each region could reacts to the sustainability and development challenges.

2. In the introduction section, the narrative conducts the reader to the importance of two groups of stakeholders in Bolivian (Guarayos indigenous communities) and Brazilian Amazon (Tapajós riverine communities). An important point in this kind of modeling approach is the choice of stakeholders to represent the multiplicity of actors and perceptions for tackling the problem analyzed. Considering this, some questions come up:
- Do the authors think that the riverine and indigenous communities were well represented in the groups of stakeholders that participated in the workshop?
*Good point. Yes, riverine and indigenous communities were well represented in the workshops. See Table 1. In both cases (Guarayos in Bolivia and Tapajós in Brazil), key representatives of the indigenous communities (with the ability to make and to influence decisions) attended the workshops. E.g., in Guarayos (Bolivia), several representatives of the Organisation Centre of Guarayo Native People (COPNAG), which is the most powerful and influential indigenous association in the region attended the workshops. Similarly, in Tapajós (Brazil), the representative of all indigenous communities of the Flona (who lived in Communidade do Maguari) attended the workshop, together with other indigenous community heads. Indigenous communities were reached by the local teams of the ROBIN project (IBIF in Bolivia, and EMBRAPA in Brazil; researchers of both teams are co-authors*

*of the paper), which are great connoisseurs in the area and have long experience working with indigenous communities.*

- Do the cognitive maps represent the vision of these groups?

*Yes. The maps include the vision of these groups. In fact, in Brazil, the representative of all indigenous communities of the Flona presented the FCM obtained in the plenary*

3. The description of the study area is long. The authors could be more focused on providing elements to support the research questions and the results (especially, the scenarios). For example, the social, cultural and political contexts experienced by stakeholders that can influence the networks structures or different responses to scenarios.

*Agree, thank you. Following your suggestion (and other similar from other reviewers) we have deleted irrelevant information (e.g., mean annual temperatures or precipitation; we have been told that it is sufficient to know the type of climate for the purpose of the article) and focused this section on the description of the socio-economic, cultural, and political context. In addition, as suggested by one of the reviewers, the description of the case study has been now separated from the methodology section. We have added a new point '2 Description of the study area' that goes after '1 Introduction' and before '3 Methodology'.*

4. Regarding the description of the study area, details of temperature, precipitation and vegetation are not relevant in this section, unless they are used in the design of climate change scenarios (that would be interesting).

*We have deleted this information. See previous comment (point 3).*

5. The section 2 should focus a little more on describing the workshops. Given that the stakeholder participants within each case study seem to be diverse and present even contrasting view on development and conservation, some issues need to be clarified, such as:

*Thank you very much for noticing this. We have made some changes following your suggestions (see below)*

- How the authors selected the stakeholders groups?

*Agree. We have clarified this. See page 5, lines 25-31 and Table 1*

How conducted the process of identifying the components to be included in the model? How do the participants identify the degree of influence between components (high, medium, etc.)?

*Agree. This has been explained in more detail in the manuscript. See page 6, lines 1-10*

What were the most important components mentioned in the workshops?

*The most important components are those reflected in the FCMs, and particularly those with the highest page rank (see Figures 4 and 5).*

- It is unclear how component values were obtained during the workshops (were individual or group responses?).

*Agree. They were group responses. This is now specified on page 6, lines 1-2 'the FCM developed represented stakeholder group knowledge' (Ösezmi and Ösezmi, 2004), and on page 6 line 11, FCMs are 'group maps'.*

How have the authors converted the cognitive maps in the adjacent matrix? I mean, how the strength of the interactions among components (the weighted values) was defined?

*The strength of the interactions among components was defined by the stakeholders in the workshops as described on page 6, lines 1-10*

How the contrasting view of the problem was converted in a single value of influence? I suggest the authors to provide the ranges of model parameters/variables presented during the workshops to show contrasting views.

*The FCMs are group maps and therefore 'agreed' or 'consensual' maps developed during the workshops. Discussions between stakeholders were guided by a facilitator, who helped to reach consensus. These types of exercises are not meant to identify contrasting views, to do so it is better to develop individual FCMs or other methodologies. Furthermore, the objective of the paper was not to dig on individual/contrasting views, but to have a clear picture of the common vision of the present in two communities (Guarayos and Tapajos) from different countries (Bolivia and Brazil), living on the edge of the agricultural frontier and confronting similar problems.*

6. Scenario section (2.4) is not clear. The authors could provide more details about the scenario conception and the stakeholders' contribution.

*Agree. Thank you. We have made some changes following your suggestions (see below)*

- How were climate change identified by stakeholders (changes in temperature, extreme climate events, precipitation, river level, floods, forest fires, soil erosion, etc.) and how were they translated it to the model?

*As is common with the Fuzzy Cognitive Mapping method, similar components were grouped together. The stakeholders in both Brazil and Bolivia mentioned an array or terms (e.g. increasing drought, reduced rains, increasing floods, weather instability), but they decided during the workshops to use the word 'climate change' to catch all terms. Further, in follow up meetings (Varela-Ortega et al., 2014) stakeholders validated this combination as being accurate to the current situation.*

*Following the suggestion of one of the reviewers, we have removed the climate change scenario as being an independent scenario and just defined it as an additional element for the scenarios. Also, following your comment, we have now provided a more extended explanation of the translation of these scenarios into the model (Pg 8: lines 7-18).*

The climate change scenarios are the same for the two study sites?
*Yes*
What climate changes were considered to be impacted by deforestation?
*Increasing drought, reduced rains, increasing floods, weather instability*

- Conservation strategies were resumed in one strategy in the Tapajós case study (Environmental Monitoring). It is not clear if the scenario components were defined in the workshop by the stakeholders or by the authors. Anyway, I see as a problem reducing conservation strategies in a unique and passive action of monitoring. By doing this, conservation strategies may seem to have low impact to achieve desired changes, in comparison with the governance and techno-social reform.

*The scenarios were first proposed by the authors, based on literature review, and then further defined by the stakeholders taking into account the limited number of factors included in the FCMs. We agree that the conservation scenario in the Tapajós case study may seem to be too reductionist, but stakeholders identified improved monitoring as the key environmental aspect to achieve a successful conservationist future. Stakeholders think that many conservation policies have already been developed and put in place, but their effectiveness has been limited due to insufficient monitoring and enforcement. Also, stakeholders think that a lot more remains to be done for improving institutional and governance systems, to protect traditional communities, support technical training, etc. Many aspects could be improved in this regard that could have positive impacts in the region. This is why the governance and techno-social scenario include changes in several components and the conservation scenario only in one.*

7. The authors show in figure 2 that FCM was validated in the second workshop. How the validation procedure was carried out? Can the same participants in the first workshop validate the FCM they created themselves?

*The people who participated in the second workshop are not exactly the same as those who participated in the first workshop. We were very careful to count with the same group of stakeholder, but the key representatives varied in some cases (due to agenda issues or changes in governmental bodies). Thus, the FCMs were validated by the same groups of stakeholders, not exactly by the same participants.*

*In the second workshop, the validation was performed by showing the stakeholders the processed FCM, including the dynamic analysis, and discussing with them the results. In both cases, Guarayos in Bolivia and Tapajos in Brazil, the main components of the FCMs remained unchanged, but stakeholders decided to change (increase/decrease) the strength of some links among components (e.g., in Guarayos, stakeholders decided to increase the weight given to the links 'illegal mining → soil erosion';  'illegal mining → contamination').*

8. In the Dynamic analysis of FCM (3.2), some interesting results could be presented in respect to the model dynamics during the calculation to achieve the baseline situation. Does the system's identity remain the same after steady state analyses is conducted?

*Yes, the systems' identity remains the same. The steady state analysis considers the current situation of all variables. It is used to measure how a variable is changing (increasing, reducing, or stable based upon the value) in the system and you can also compare across variables (i.e whether deforestation is increasing, whilst forest law implementation is reducing) within the system. However, the system remains the same as the weighting applied to each variable is identical; the 'identity' would only change with the application of the scenarios, where the current situation of the system is altered. The iterations (calculation) of the model dynamics are irrelevant, the final result is what it is important and it is shown in Figures 6 and 7.*

- Do the authors think that there is a relation between FCM complexity and the diversity of indigenous and riverine communities in the group of stakeholders?

*No, we think that there is not such a relation. Key representatives transmitted a common voice for the indigenous and riverine communities. This is quite frequent; they used to have a common voice to make theirselves heard.*

Specific comments:

Page 2: Include more recent citations in introduction.
*Agree. Done*

Page 5, Line 6: 'dense moist and wet forest types'. I suggest you include a classification system for the Amazon vegetation to describe the forest types.
*Agree, defined now as dense terra firme (upland) tropical moist forest*

Page 5, lines 7-9: I suggest you mention the decree n_ 73.684, February 19 of 1974.
*Agree, included*

Page 5, Line 10: Cite data source (reference) in respect to the number of 'ribeirinhos' and 16 communities mostly along Tapajós river.
*Following the suggestion of another reviewer, we have deleted the number of communities and further specified the ethnical background of this riberirinhos*

Page 6, Lines 20 – 25: How were focus groups defined?
*This has now been detailed in Pg. 5, lines 26-31.*

Page 7: Last paragraph: I suggest to present the adjacent matrix in the supplementary material.
*Fist response (November 2019)→ We have tried to include the adjacency matrix as Tables in the Supplementary material, but it has been impossible, they are too big (29 lines x 29 columns in Guarayos, 32 lines x 32 columns in Tapajós). They are illegible*
*According to the Editor's comments (December 2019) → We have included the 2 adjacency matrix (S1 Guarayos and S2 Tapajos) in the Supplementary Material.*

Page 9: How much components were included in the model by the workshop participants?
*This is specified in Table 4 (second line), in Guarayos 29, in Tapajós 32*

6-  Editor

**Editor Decision: Publish subject to minor revisions (review by editor)** (14 Dec 2019) by Wenwu Zhao
Comments to the Author:
It is a good work. While, I'd like to give some additional suggestions before acceptance.
*Thank you very much for the review and good suggestions.*

1.Introduction. Page2, Line1. Please provide references for "Amazon basin is the world's richest biological reservoir and a globally significant carbon sink".
*Thank you. Done (see Page 2, Lines 2-3) → WE REFER TO THE 'CLEAN' DOCUMENT*
2.Introduction. Page2, Lines 5-6. Please provide references for "Weak governments and political instability in Amazonian countries have reduced capacity to halt deforestation and related expansion of illegal activities".
*Thank you. Done (see Page 2, Lines 6-7)*
3.Introduction. Page 2, Lines 7-8. This sentence may not necessary for this paper.

*We are not sure if you refer to the 'megafires' sentence or the 'climate change' sentence. The megafires sentence was asked by one of the reviewers, so we have kept it. With respect to the climate change sentence, following your suggestion, we have deleted it and merged it with the last sentence of the paragraph (see next point). See Page 2, Lines 7-10*

4. Introduction. Page 2, Lines 9-10. There are some ambiguous expressions on future scenarios, please give detail description on the consequences of climate change you described in the previous sentences.

*Agree. We have rephrased this sentence:*
*'Future scenarios depict a reduction in tree coverage and increased drought in the Amazonia (e.g. Guimberteau et al., 2017; Malhi et al., 2008; Tejada et al., 2016), with Lenton (2011) proposing that ecological tipping points could be reached.' See Page 2, Lines 8-10*

5.Introduction. Page 2, line 24-32, You have listed several case studies here "Nobre et al. (2016) promote a "third-way", driven by investment in technical and social capital, catalysing a localised industrial revolution. Guedes et al. (2014) offer that increased access to technical assistance may permit communities to develop more sustainable livelihoods, converting natural capital to social. Lapola et al. (2014) infer that technological improvements along with sustainable land management could drive sustainable land use shifts. A further possibility may lie in the results of recent analyses, which suggest that socio-economic development in forest frontier regions of Brazil has uncoupled from environmental exploitation and degradation, due to policy development and implementation (e.g. Weinhold et al., 2015; Caviglia-Harris et al., 2016).Tritsch and Arvor (2016) propose that recent improved governance structures have begun to address competing rural development goals. Godfray et al. (2011) and Newton et al. (2013) advocate that governance and institutional improvements could provide a balance between conservation, development, and climate change mitigation". Could you find some better way to list them? For example, is it possible for you to categorize these examples and condense them in a clearer way?

*Thank you. We have structured this paragraph more clearly. Nobre et al. (2016) says that investing in technical and social capital is key for promoting sustainability (as a 'third-way'). So, we have put first all the studies that talk about improving the technical aspect, and then those that talk about the importance of investing in social capital. See Page 2, Lines 22-33.*

6.Introduction. Page 3, Line 1 and Line 5. The two "However" are not well readable, please reconstruct these sentences. It may be better to introduce the significance of stakeholders' relevant studies firstly, then propose the necessity on conducting stakeholders' evaluation in this region.

*Thank you very much for this remark. Following your suggestion, we have reconstructed these sentences. First, we have explained the potential benefits of engaging the stakeholders in scenario analysis and second, we have explained the added value of doing it in the Amazon region. See Page 3, lines 6-17.*

7.Introduction. Page 3, Lines 13-19. This paragraph is most important, but it is not well enough. You'd better to arrange the specific aims in accordance with order of the previous introduction.
*See our previous comment*

8.Page 4 line 15, What's meaning of IUCN? Please give its explanation.

*Agree. Done. We have clarified that it refers to 'International Union for Conservation of Nature'. It refers to a well-known list of protected area categories. See Page 4, Lines 16-17.*

9.Methodology. Section 3.1. This section has several minor shortages. For example, Page5, Lines 11-14 has three discontinuous references, which may not readable.
*Thank you for the observation. It has been corrected. See Page 5, Lines 11-14.*
Lines 14-19 are all related to function of FCM, which should be reduced by one or two concise sentences.
*Agree. Done. We have left two sentences: 1- related to 'participatory' FCMs, and 2- related to FCMs (at large, not necessarily participatory). See Page 5, Lines 14-17.*
Page 6, Lines 2-3 may not logical.
*We first asked the stakeholders what was the current state of the environment. Then, we asked them what they would do to improve the described state. The sentence is logical, but may be not necessary (repetitive with the text that follows), we can delete it if you prefer so.*

10. Results. Page 9, Lines 6-10. These sentence not only for results, but combine some discussion sentences, such as "The density difference may suggest that stakeholders in Tapajós perceive greater causal relationships between components". It may be necessary to proofread the whole results and do revisions wherever needed.
*Thank you. The entire paragraph has been re-written to avoid confusion. See Page 9, Lines 1-6. We have checked the rest of the text and we think that it is fine.*

11.Discussion. I am not sure the sub-title of Section 5.1 is good. It may be better to summarize the discussion and replace by other appropriate title.
*Thank you for this remark. We have summarized the first paragraph, and changed the sub-title of Section 5.1, now 'The current picture of the Amazon', to highlight that we are discussing here the present situation and in the next sub-section 5.2 the future situation. See Page 12, Lines 19-25.*

12.Discussion. Page 13, Line 9. The word "who" may be replaced by "which".
*Agree. Done. See Page 13, Line 1.*

13.Discussion. Page 13, Line 12. Complement further explanation of the causes on difference.
*Agree. It is now better explained: 'In particular in Tapajós, climate change impacts are associated with an increase in extreme weather events (heat weaves, droughts) and soil dryness; that is because they are considered a main cause of the wildfires occurrence in the region. Varela-Ortega (2014) found that stakeholders perceived climate change a fundamental component in the future of both regions and in Tapajós in the present.' See Page 13, Lines 3-6.*

14.Discussion. Page 15. Line 11. In the previous sentences, you state the features of both institutional reforms and techno-social reforms, then you propose the issue of investment as well as the realistic status. These are critical and undoubtful. While, you may also need to list some solutions for these according to the results of your research. Current content is not enough.
*Done. we have added the following text:*

*'Our results show that some techno-social measures, such as improving environmental awareness, may be effective to promote sustainability, while do not necessary require a large amount of public financial investments. Actions of social responsibility can contribute to financing investments in education and new technologies. Also, governance measures (e.g., better coordination of laws and institutions) can facilitate the adoption of more ambitious techno-social measures (e.g., better training and assistance, protection of traditional communities) at lower cost. Thus, further analysis should be performed on synergies between governance and techno-social measures, as well as on collective work between the public and the private sector to better organize and prioritize investments and actions.'*
*See Page 15, Lines 6-12.*

15. As for the comments "Page 2: Include more recent citations in introduction", please list the references you have been added in the Author Response.
*We have included 4 recent citations in the introduction:*

- *Garcia, A., Vilela, VM de F.N, Rizzo, R., West, P.C., Gerber, J.S., Engstrom, P.M., Ballester, M.V. (2019). Assessing land use/cover dynamics and exploring drivers in the Amazon's arc of deforestation through a hierarchical, multi-scale and multi-temporal classification approach, Remote Sensing Applications: Society and Environment, 15, 100233, https://doi.org/10.1016/j.rsase.2019.05.002*

- *Guimberteau, M., Ciais, P., Ducharne, A., Boisier, J.P., Dutra Aguiar, A.P., Biemans, H., De Deurwaerder, H., Galbraith, D., Kruijt, B., Langerwisch, F., Poveda, G., Rammig, A., Rodriguez, D.A., Tejada, G., Thonicke, K., Von Randow, C., Von Randow, R.C.S., Zhang, K., Verbeeck, H. (2017). Impacts of future deforestation and climate change on the dydrology of the Amazon Basin: a multi-model analysis with a new set of land-cover change scenarios, Hydrol Earth Syst Sci, 21, 1455-1475, https://doi.org/10.5194/hess-21-1455-2017*

- *Global Forest Watch. (2019). Available at https://www.globalforestwatch.org/map, Last Accessed: 17/09/19, 2019.*

- *Zemp, D.C., Schleussner, C.-F., Barbosa, H.M.J., Rammig, A. (2017). Deforestation effects on Amazon forest resilience. Geophys Res Lett, 44, 6182-6190, https://doi.org/10.1002/2017GL072955*

16. SDGs is one of the most hot word for sustainability. Could you give more general background on sustainability? Such as SDGs. It may help to contact your research with UN targets, and have a more important significance.
*Thank you for addressing this issue. This is a very interesting comment. In fact, we have a second paper ready for submission, which is a continuation of this one and focuses on SDGs. We look at the future (with FCM based on RCP and SSP scenarios) and study priorities for action in the context of SDGs. We prefer to keep this paper as it is and discuss the SDGs issues in the second paper.*

17. Figure 9. Complement full name of Agri. Development in the caption.
*Agree. Done. The full name has been added to Figure 8, S1 and S2 too. Minor errors were found in F8 and S1, and they have been corrected now.*

18.Figure 3. Delete the "%" in labels of y axis.
*Agree. Done.*

19.Figure 1. Complement legend for the Amazon Basin.
*All legends are explained below the title (see e.g. Figure 4, Figure 5…). We have done the same here.*

20.Table 2. This table is better to be positioned in supplemental materials.
*Agree. Done.*

[revised manuscript text omitted]